# Predicting the spatiotemporal diversity of seizure propagation and termination in human focal epilepsy

Timothée Proix[1,2,3], Viktor K. Jirsa[4], Fabrice Bartolomei[4], Maxime Guye[5] & Wilson Truccolo[1,2,3]

Recent studies have shown that seizures can spread and terminate across brain areas via a rich diversity of spatiotemporal patterns. In particular, while the location of the seizure onset area is usually invariant across seizures in an individual patient, the source of traveling (2–3 Hz) spike-and-wave discharges during seizures can either move with the slower propagating ictal wavefront or remain stationary at the seizure onset area. Furthermore, although many focal seizures terminate synchronously across brain areas, some evolve into distinct ictal clusters and terminate asynchronously. Here, we introduce a unifying perspective based on a new neural field model of epileptic seizure dynamics. Two main mechanisms, the co-existence of wave propagation in excitable media and coupled-oscillator dynamics, together with the interaction of multiple time scales, account for the reported diversity. We confirm our predictions in seizures and tractography data obtained from patients with pharmacologically resistant epilepsy. Our results contribute toward patient-specific seizure modeling.

[1] Department of Neuroscience, Brown University, Providence, RI 02912, USA. [2] Institute for Brain Science, Brown University, Providence, RI 02912, USA. [3] Center for Neurorestoration & Neurotechnology, U.S. Department of Veterans Affairs, Providence, RI 02912, USA. [4] Institut de Neurosciences des Systèmes (INS), Inserm, Aix Marseille Univ, Marseille 13005, France. [5] CNRS, CRMBM UMR 7339, Aix Marseille Univ, Marseille 13005, France. Correspondence and requests for materials should be addressed to W.T. (email: wilson_truccolo@brown.edu)

Brain dynamics during epileptic seizures present various micro- and macroscopic spatiotemporal structures at different seizure stages. For patients with drug-resistant epilepsies, better understanding the mechanisms underlying these spatiotemporal patterns during seizure initiation, propagation, and termination is crucial to improve treatment methods. These could include resective surgery of seizure onset zones[1–3] and new therapeutic approaches based on seizure prediction and abatement by electrical stimulation[4–7].

Understanding and modeling the complex mechanisms underlying the many aspects of seizure spatiotemporal dynamics is a difficult task, as different spatial and temporal scales interact. In addition, as new recording technologies have become available and studies have provided more detailed description of the dynamics of seizure propagation, maintenance, and termination, a diversity of apparently contradictory phenomena has emerged. Different models have tackled some of these phenomena, such as the spatiotemporal dynamics of seizure propagation[3,8–10], the source and direction of spike-and-wave discharges (SWDs) during seizures[8,11,12], or the mechanisms supporting seizure onset and offset[13,14]. Here, we consider focal seizures that start with or evolve into SWDs[15]. SWDs during focal seizures are characterized by a large-amplitude spike in the field potential followed by a slower wave with the opposite polarity. More complex morphology, e.g., poly-spikes, is also common. SWD events tend to recur 2–3 times per second during a seizure. Typically, neuronal action-potential activity increases during the spike phase and is largely suppressed during the wave phase[16,17]. We focus on two main related aspects of these SWD seizures. First, while seizures propagate slowly to connected areas with speeds on the order of 1 mm/s[17,18], ictal waves in the form of SWDs propagate orders of magnitude faster (100–1000 mm/s)[8,12,19]. Furthermore, recent studies report apparently contradictory results regarding the source of fast-propagating SWDs. Smith et al.[12] report that the slow ictal wavefront is the moving source of SWDs. In contrast, a study by Martinet et al.[8] supports that the source driving the fast SWDs remains stationary at the initial seizure onset area. Second, seizure termination has long been characterized as a synchronous (relatively to the slow propagation time) event across the recruited brain areas[8,12,13]. It is clear, however, that not all seizures terminate synchronously across the brain. In many cases, seizures may evolve into different clusters or regions of activity[9], with ictal activity in each cluster terminating synchronously, but with long termination delays between the clusters spanning tens of seconds.

In this article we introduce a unifying neural field model that explains the diversity of previously observed phenomena including seizure initiation, propagation across local and large-scale brain networks, and termination. To do so, we extend the Epileptor neural mass model[14], which is a canonical model for seizure temporal dynamics, into a neural field model. Neural fields are based on space and time coarse-graining of synaptic or firing rate activity in neuronal populations[20] to capture the spatiotemporal evolution of collective state variables at the population level. The novel Epileptor field equations incorporate homogeneous (local) short-range and heterogeneous long-range connectivity. The choice of homogeneous connectivity can be informed by previous anatomical and physiological studies (e.g., see refs. [21,22].). Heterogeneous connectivity can be derived from diffusion magnetic resonance imaging (MRI) to build whole-brain models, e.g., see ref. [3]. Two main properties of the Epileptor field model allow us to predict and account for the controversial findings regarding seizure spread and termination described above. First, the propagation of the slow ictal wavefront and fast SWDs are supported by two different mechanisms: wavefronts in excitable neural media and coupled-oscillator dynamics,

respectively[23]. While changing the seizure onset area requires changing the model parameters, the source of the SWDs can dynamically move or remain stationary in the initial seizure onset area, depending on the neural field excitability and on coupled-oscillator phase reorganization during the seizure. Second, although the slow seizure propagation and the faster SWDs are supported by two different mechanisms, we show that they are intrinsically related. Specifically, while the dynamics of wave propagation in excitable neural fields determine, in part, the source and direction of SWDs, the latter contributes to drive the types of spatiotemporal patterns during seizure termination, without requiring the intervention of a global parameter change. In addition, we also show that (>10 Hz) low-voltage fast-activity (LVFA), resulting from the multiple time scale interactions in the neural field model, can hamper the propagation of the slow ictal wavefront. These dynamics, together with variations in short- and long-range connectivity strength, play a central role in seizure spread, maintenance and termination. We demonstrate our predictions in a cohort of 13 epileptic patients, using stereotactic electroencephalographic (SEEG) recordings and patient-specific tractography obtained from diffusion MRI.

## Results

**Epileptor field model.** We introduce a new neural field model to account for observed local field potential (LFP) spatiotemporal dynamics during seizure initiation, propagation and termination (Fig. 1). The new model is obtained via a field extension of the Epileptor neural mass model, originally formulated for the LFP temporal dynamics in focal epileptic seizures[14]. To extend the Epileptor neural mass model to a neural field model, we

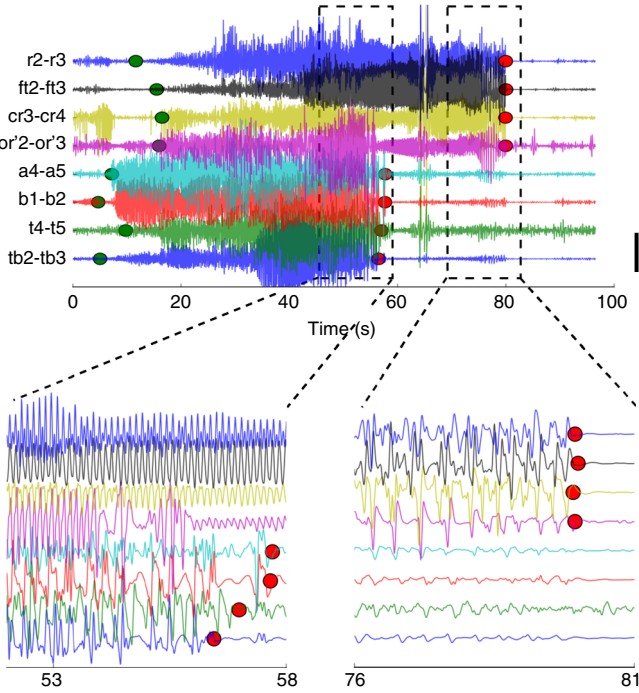

**Fig. 1** Example of a seizure recorded via stereotactic EEG electrodes. The top plot shows the full extent of the seizure. Green (red) points indicate seizure channel-wise onset (offset) as determined by visual inspection. Large delays between different brain areas can be observed between different channels both for seizure onset and offset. Scale bar: 1 mV. The bottom left (right) plot magnifies the seizure termination for the first (second) cluster of recorded brain areas. Channels where the seizure ended simultaneously show coherent spike-and-wave activity. Colors denote the corresponding channels in the top and bottom plots

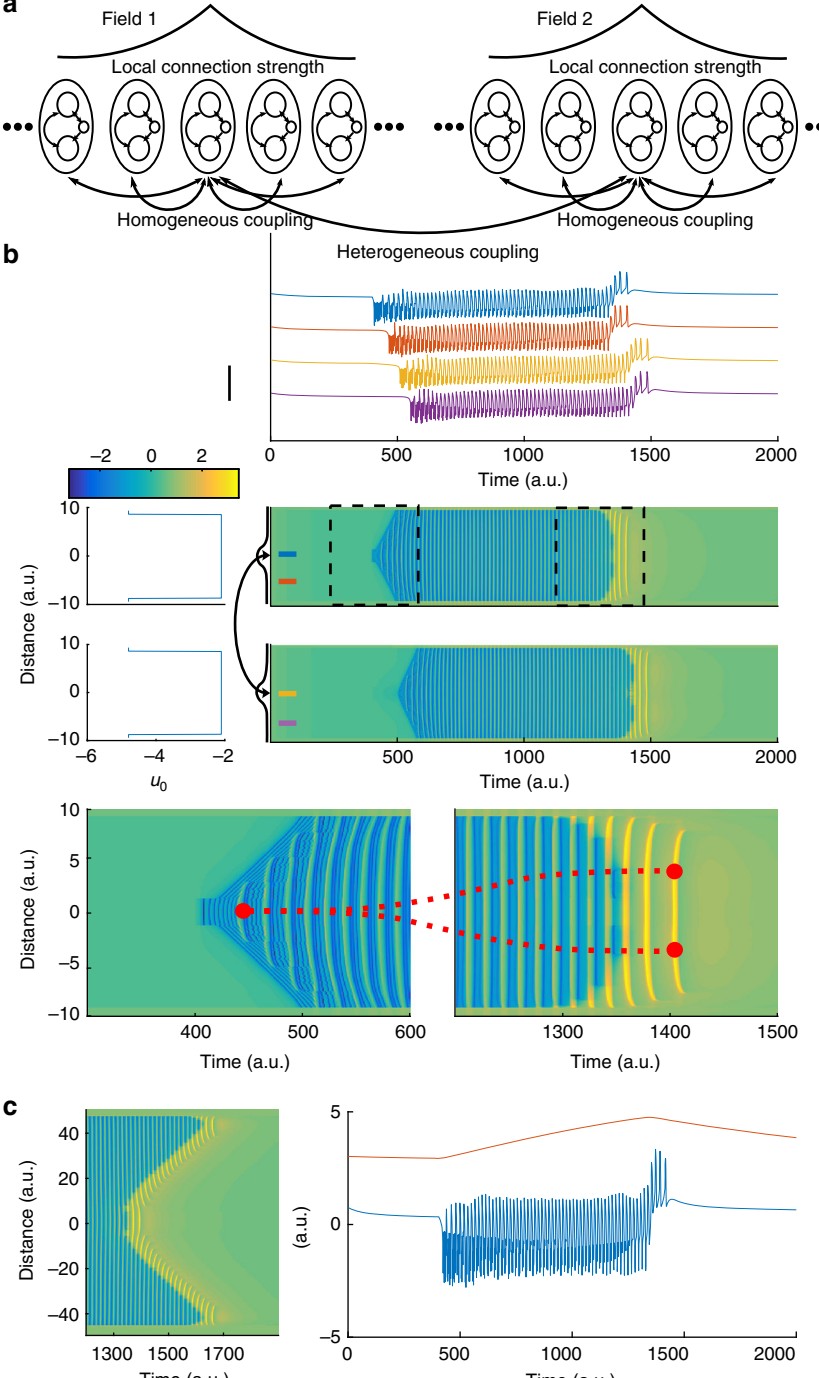

**Fig. 2** The Epileptor field model reproduces multiscale features of spatiotemporal seizure dynamics. **a** Schematics for the Epileptor field model. At each spatial location of the neural field, an Epileptor site (oval) includes two neural populations and a slow permittivity variable (inner circles). Epileptors are connected to their neighbors via local homogeneous coupling, which decays exponentially with pairwise distances. Epileptors also connect to Epileptors in other fields through heterogeneous coupling. **b** Simulation of a model including two distant Epileptor fields (middle plot). The top field includes the seizure onset area, while the bottom field represents a more distal brain area. Spatial values for excitability parameter ($u_O$) and simulated spatiotemporal activity are shown on the left and right, respectively. For simplicity, we chose a constant spatial profile for $u_O$, except for the regions close to the field boundaries, which represent brain regions with low excitability that are not recruited into the seizure. The curved arrow indicates heterogeneous connections between the two fields. The connection kernels (black lines) were centered at the indicated locations. The seizure started in the top Epileptor field, slowly propagated throughout the field, then eventually recruited the bottom Epileptor field, where it slowly propagated as well. The horizontal colorbar indicates the activity level for the middle and lower plots. Time series at four spatial locations in the two fields (indicated by colored bars in the middle plot) are shown in the top plot. Scale bar: 4 a.u. The two bottom plots show a zoomed in view of the spatiotemporal activity at seizure onset (left) and offset (right). In this example, the seizure ended synchronously within each field, with a few SWDs propagating after seizure offset. Red dots mark the source location of the propagating SWDs at seizure onset and offset. As the seizure evolved, the source of SWDs changed along with the slow ictal wavefront across each field. **c** Left: Asynchronous seizure termination; in this case the recruitment delay was large enough in comparison to the seizure duration; Right: Corresponding activity ($-u_1 + q_1$, blue) and the slow permittivity variable ($v$, red) at the seizure onset site (location zero in the field)

introduced a spatially homogeneous local coupling between neural populations using spatial convolutions with local connectivity kernels (see Fig. 2a and Methods). A detailed justification for the choice of such coupling kernels and functions is provided in the Methods section. While in the Epileptor mass model the state variables are a function of time ($t$) only, in the Epileptor field model they are functions of both time ($t$) and space ($x$). The local coupling function was chosen to be a Heaviside firing rate function. To model seizure dynamics across distal brain areas, we examined a set of two neural fields coupled via long-range heterogeneous connectivity and coupling kernel (see Methods). Extensive exploration of the coupling parameter space showed that the qualitative dynamics presented here are robust over a large range of these parameter values (Supplementary Fig. 1).

As in the original mass model, the Epileptor field model includes neural dynamics with three different time scales related to: (i) the transition between interictal and ictal periods (carried by slow changes in a variable $v(t)$), (ii) the emergence of LVFA oscillations (carried by a first neural population on a fast time-scale, variables $u_1(t)$ and $u_2(t)$), and (iii) 2–3 Hz SWDs (carried by a second population on an intermediate time-scale, variables $q_1(t)$ and $q_2(t)$). The fast and intermediate time scales interact via homogeneous connectivity in the field extension. The slow variable $v(t)$ is termed the "permittivity" variable, as it represents the distance of the Epileptor state to the seizure threshold, i.e., the ability of the model to resist to seizure triggering events. The slow permittivity variable captures slowly evolving physiological processes in the brain, such as changes in the extracellular concentration of different ions[24–28], metabolism[29,30], and tissue oxygenation[31,32]. The Epileptor's ability to transition into seizures autonomously is mediated by an excitability parameter ($u_0$). For $u_0 > -2.1$, the Epileptor triggers seizures autonomously and is said to be epileptogenic. Although the Epileptor field model can spontaneously transition in and out of seizures (Supplementary Fig. 2), for convenience, we here trigger seizures through local stimulation at the center of one of the Epileptor fields. Figure 2b shows a simulation of the two connected Epileptor field models. Once initiated, the seizure slowly propagates through an expanding wavefront recruiting adjacent areas from the interictal into the ictal state. In turn, the second Epileptor field is recruited after a longer delay, displaying a similar propagating ictal wavefront. As the seizure evolves, SWDs emerge, propagating with a much higher speed than the slow ictal wavefront. Importantly, the origin and directions of the propagating SWDs can either change through the seizure (Fig. 2b), or alternatively, the SWD sources may remain spatially stationary, as shown below. Finally, a seizure can either terminate synchronously or asynchronulsy (Fig. 2c). As in actual seizures (Fig. 1), there can be a substantial termination delay between the seizure offset times between the two fields. In the next sections, we demonstrate how the proposed Epileptor field model accounts for the diversity in spatiotemporal dynamics during seizure spread, maintenance and termination.

**Ictal wavefront propagation in excitable neural media.** When a seizure initiates in a localized region of the field, this region excites adjacent non-recruited territories; a slow traveling wavefront solution is formed (Fig. 3a) and propagates in the excitable medium. The existence of this slow propagating wavefront solution depends on the local coupling in the fast variable $u_1$ of the Epileptor field model (Eq. 1 in Methods). The mechanism of this front propagation can be visualized in a phase space plot representing the fast variables $u_1$ and $u_2$ (Fig. 3b). As a seizure starts in a region, the local coupling function will excite the neighboring

Epileptors from a stable interictal state into an oscillatory ictal state. The ictal wavefront solution is thus generated by the propagation of the excited activity from the stable fixed point to the oscillatory activity of the limit cycle (Fig. 3b, blue lines). Importantly, the slow wavefront solution exists even when the permittivity variable ($v(x,t)$) is constant throughout the Epileptor field model. We also note, as shown in Fig. 2, that a stage-like recruitment from one brain region to another with large delays can be obtained as well by introducing heterogeneous connections. This will be discussed in more detail in the next sections.

**Fast oscillations at onset hamper seizure propagation.** To identify the mechanisms constraining the propagation speed of ictal wavefronts, we reduced the Epileptor field model to a two-dimensional system ($u_1,u_2$) using averaging methods[33]. We obtained the speed of the ictal wavefront by using a semi-analytical method known as the shooting method (Methods and Supplementary Fig. 3). The speed of propagation of the front solution was computed as a function of the local coupling strength ($\gamma_{11}$) and the threshold of activation ($\theta_{11}$) for the Heaviside firing rate function in the Epileptor field model (Eq. 1 in Methods, Fig. 3c, d) for different values of the permittivity variable ($v$). As shown in Supplementary Fig. 4, the semi-analytical shooting method and the numerical simulations of the full system led to consistent results. To correctly evaluate the speed of propagation, the shooting method must include the two variables of the fast population ($u_1$ and $u_2$), in order to allow for fast oscillations (LVFA) to emerge on the ictal wavefront. The way these fast oscillations affect the ictal wavefront speed can be understood as follows. When fast oscillatory activity emerges during the ictal state at a given site, this oscillatory activity moves "up and down" the operating point of the nonlinear coupling function. In the particular case of a Heaviside coupling function, when the oscillatory activity is below the threshold of the function, the output is zero. Thus, the fast oscillatory activity modulates the effective coupling. Because the propagation speed of the ictal wavefront is dependent on the effective coupling strength, the net effect of fast oscillations is to reduce the propagation speed. Conversely, when fast oscillatory activity is removed from the model, the operational point of the Heaviside function at the ictal sites remains above the threshold, resulting in sustained and high effective coupling. We note that the removal of fast oscillations was obtained by model reduction via averaging of the activity of $u_2$ over time[33]. Without fast oscillations, the ictal wavefront speed is then one order of magnitude higher than without averaging (Methods and Supplementary Fig. 5). Thus, fast oscillations are a critical element in slowing down the speed of propagation in the Epileptor field model. This mechanism remains valid for other nonlinear functions such as sigmoidal firing-rate functions. Although LVFA oscillations have been associated with inhibitory activity at seizure onset[28,34] and are consistent with the inhibitory veto hypothesis[17,35], our model points to a more general dynamical mechanism for how LVFA oscillations can effectively hinder seizure propagation. The dynamical mechanism is more general in the sense that other sources of LVFA oscillations, not necessarily requiring the intervention of surround inhibition (inhibitory veto), can lead to the same hampering effect.

**Coupled-oscillator dynamics of fast propagating SWDs.** The phenomenon of slow ictal wavefront propagation examined above can be understood as wave propagation in excitable media. In contrast, the transition into oscillatory states characterized as ictal SWDs brings additional features belonging to coupled-oscillator systems. Ictal SWDs emerge during the seizure and

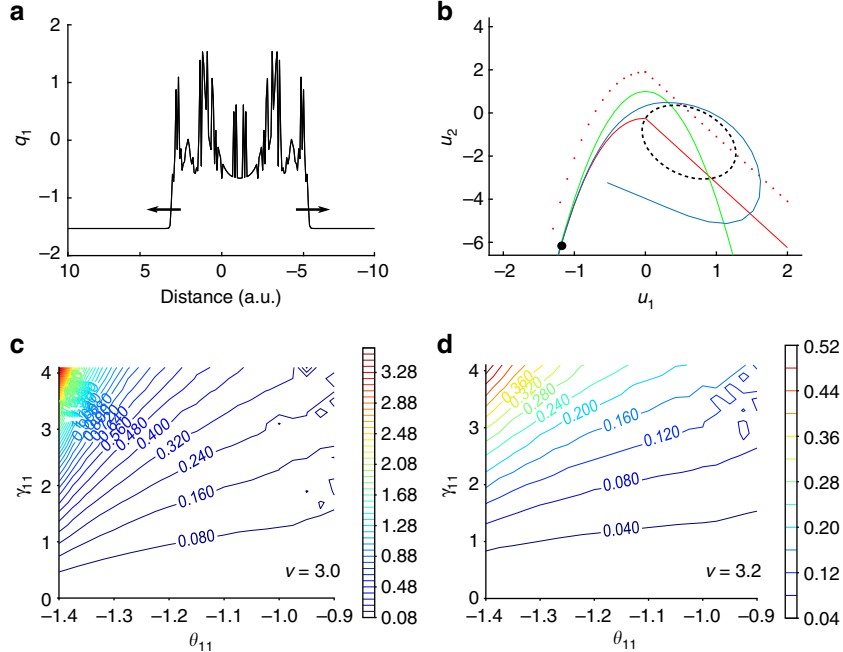

**Fig. 3** Seizure propagation via slow-traveling ictal wavefront in the Epileptor field model. **a** Plot of a traveling front solution in the fast population. **b** Phase space for the fast population variables ($u_1$ and $u_2$). In the interictal state, the system is on the left stable fixed point (black dot) at the crossing of the cubic-linear nullcline $\dot{u}_1 = 0$ (full red line) of Eq. 1 and the squared nullcline $\dot{u}_2 = 0$ (green line) of Eq. 2 (Methods). After seizure onset, excitation through local coupling moves the red nullcline of neighboring Epileptors upward (dashed red line) until disappearance of the left stable fixed point, resulting in the activity (blue line) to jump to the stable limit cycle (black dashed line). **c** Speed of propagation (a.u.) of the ictal wavefront as a function of the activation threshold $\theta_{11}$ and the connection strength $\gamma_{11}$, with the permittivity variable set to $v = 3.0$. The speed of propagation is slow if the threshold of activation $\theta_{11}$ is higher than the value of the fast population activity ($u_1$) in the interictal state. **d** Same conventions as in **c** for a permittivity variable $v = 3.2$. The speed of propagation remains slow for other values of the permittivity variable

travel rapidly (100–1000 mm/s[8,12]) in comparison to the slow ictal wavefront (Fig. 4a). In the Epileptor field model, this is achieved by the local coupling in the second population (variable $q_1$). To understand the mechanisms underlying the fast propagation of SWDs, we used averaging methods[33] to isolate the second population from the other variables. We expressed the equations for the second population in terms of a variable $K$. The value of $K$ is constant under the averaging approximation and depends on the slow permittivity variables and the average activity of the fast population ($u_1$, Methods). The variable $K$ represents the instantaneous values of slowly evolving variables during the seizure, such as the extracellular potassium concentration. Simulations of the full model indicate that during the seizure, the variable $K$ takes values that push the second population into an oscillatory regime. The activity of the second population therefore behaves as a chain of coupled oscillators (Fig. 4b). In this case, and in contrast to the slow wavefront propagation in excitable media, the propagation of SWDs originate as phase differences across the different neural oscillators instantiated by the second population in the neural field model. As we show in the next sections, coupled-oscillator dynamics support arbitrarily fast SWD propagation and rapid movement of the spatial source of the SWDs, as opposed to wave propagation in excitable media.

We quantified the SWD propagation speed and compared it with the propagation speed of the slow ictal wavefront supported by the fast population. The speed of propagation of the SWDs during the seizure depends on several parameters, such as (i) the instantaneous phase $\phi$ and the intrinsic frequency of each neural oscillator, (ii) differences in phase and intrinsic frequency among the oscillators, as well as (iii) the variable $K$ and the coupling strength $\gamma_{22}$ between oscillators. To evaluate the speed of the

SWDs, we approximated the dynamics of the neural field by the dynamics of a chain of uncoupled relaxation oscillators. This approximation was reasonable as confirmed by the numerical simulation of the full Epileptor field model (Fig. 4c, d). Next, we obtained the speed of the SWDs as a function of the variable $K$ (Fig. 4c) and initial phase $\phi_0$ (Fig. 4d), while fixing all of the other parameters (Methods). The speed values we obtained are two orders of magnitude higher than for the speed of propagation of the ictal wavefront, in agreement with previously reported experimental data. Interestingly, the coupling strength does not influence much the propagation speed (Fig. 4c, d).

**Source and direction of SWD propagation.** The source and propagation directions of ictal SWDs remain a controversial issue, with apparently contradictory data supporting either a moving source consisting of the slow ictal wavefront itself[12] or a spatially stationary source at the seizure onset area[8]. The Epileptor field model shows that both cases are possible. Specifically, the location of the source is defined as the site in the neural field that first triggers an ictal SWD during the seizure. As stated previously, the location of this initial site depends on several parameters and variables, which may evolve in different ways during different seizures. In particular, for the full Epileptor field model, the variable $K$ changes during the seizure, as shown in Fig. 5a. The maximum of this variable $K$ at a given time across the field determines the source of the next SWD. Furthermore, the excitability ($u_0$) affects the time course of $K$ (Fig. 5b). Thus, the location of the maximum of $K$, and therefore the source of the SWDs, changes during the seizure as a function of the excitability in different brain areas. In one dimension, the seizure propagates in both directions (starting at the same source as for the SWDs),

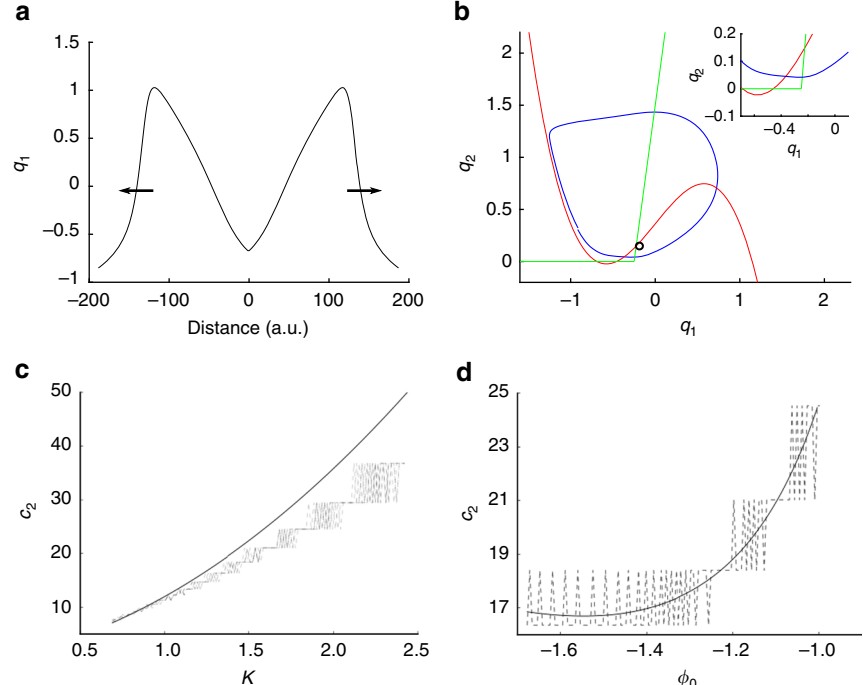

**Fig. 4** Fast traveling spike-and-wave discharges in the Epileptor field model. **a** Pulse solution for the second population (see also Methods). The spatial extension of the Epileptor field model is shown larger than in previous figures for a better visualization of the pulse solution. **b** Phase space for the second population variables ($q_1$ and $q_2$). The variable $K$, representing slowly evolving processes (e.g., extracellular potassium concentration) during the seizure, is high enough during the seizure so that the stable fixed point of the system disappears by colliding with the other unstable fixed point on the circle (saddle-node bifurcation on invariant circle), resulting in oscillatory activity (limit cycle) around the remaining unstable fixed point. Red: nullcline for the variable $q_1$ ($\dot{q}_1 = 0$). Green: nullcline for the variable $q_2$ ($\dot{q}_2 = 0$). Blue: trajectory of the pulse solution. Black circle: unstable fixed point. **c** Speed of SWDs ($c_2$) in population 2 as a function of the variable $K$, i.e., for fixed values of the permittivity variable and activity of the fast population. The speed of SWDs is two orders of magnitude higher than the slow ictal wavefront propagation and can vary substantially according to the values of variable $K$ during the seizure. The black line indicates the prediction from the analytical solution based on a chain of uncoupled relaxation oscillators, while the different dashed lines indicate the speed obtained by direct numerical integration of the full Epileptor neural field model, for different values of the coupling strengths ($\gamma_{22}$). **d** Speed of SWDs ($c_2$) in population 2 as a function of its initial phase ($\phi_0$) for $K = 1.5$. The speed of SWDs varies according to the instantaneous phase of the different oscillators. Same conventions as in **c**

until it reaches non-recruited regions or meets another SWD emitted by a different source. In sum, both moving and stationary sources are possible depending on the evolving dynamics. For example, a moving source can occur if the brain region at the origin of the seizure and the recruited areas show similar epileptogenicity (Fig. 2). On the other hand, it is also not difficult to obtain spatially stationary sources. For example, a spatially stationary source can occur if the brain region at the origin of the seizure is more epileptogenic than the recruited areas (Supplementary Fig. 2).

**SWDs in ictal clusters and synchronous seizure termination.** Traveling waves and pulses are easily obtained in neural field models and have been extensively described[20]. Similarly, for a single Epileptor field model with only homogeneous connections, one could expect that seizures would propagate as traveling pulses across brain regions. As a consequence, a slow ictal wavefront would be followed by a slowly propagating seizure offset (termination) event across the recruited areas. However, electrocorticograms (ECoGs) and SEEGs show that seizures can terminate synchronously (Fig. 1). We show here that synchronous seizure termination can occur under certain conditions across an Epileptor field model with homogeneous connections.

To model such synchronous seizure termination within a given ictal cluster, a possible solution is to vary a global parameter for the ictal cluster, which renders the front solution unstable at

seizure offset. Instead, we pursued a different mechanism that avoids the assumption of a global parameter. In the original single Epileptor neural mass model[14], the seizure ends when the fast oscillatory activity crosses the separatrix between ictal and interictal state through a homoclinic bifurcation. A homoclinic bifurcation is characterized by the disappearance of a limit cycle after its collision with a saddle point, i.e., the intersection of stable and unstable manifolds. Before seizure termination, these fast oscillations are triggered by traveling SWDs. In turn, SWDs are triggered during the seizure by the slow drive of the fast population on the second population, i.e., through the temporal convolution $g(u_1)$ acting on $q_1$ (Eq. 4 in Methods, Fig. 6a). In the Epileptor neural field model, this mechanism was extended spatially through a coupling kernel to trigger a homoclinic bifurcation (Eq. 6 in Methods). As a consequence, as the field approaches seizure termination, the Epileptors are more likely to cross the separatrix at about the same time via the propagation of a single SWD, resulting in a synchronous seizure termination (Fig. 2 right lower plot). Therefore, for a single Epileptor field model, SWD propagation just before synchronous termination (small seizure termination delays) also leads to high pairwise correlation values between LFPs at different spatial locations in the Epileptor field model (Fig. 6b). An important condition for synchronized seizure termination to work in an Epileptor field model is that the recruitment time (related to the propagation of the slow ictal wavefront) be shorter than the seizure length, so that the Epileptors are all close to seizure termination at about the

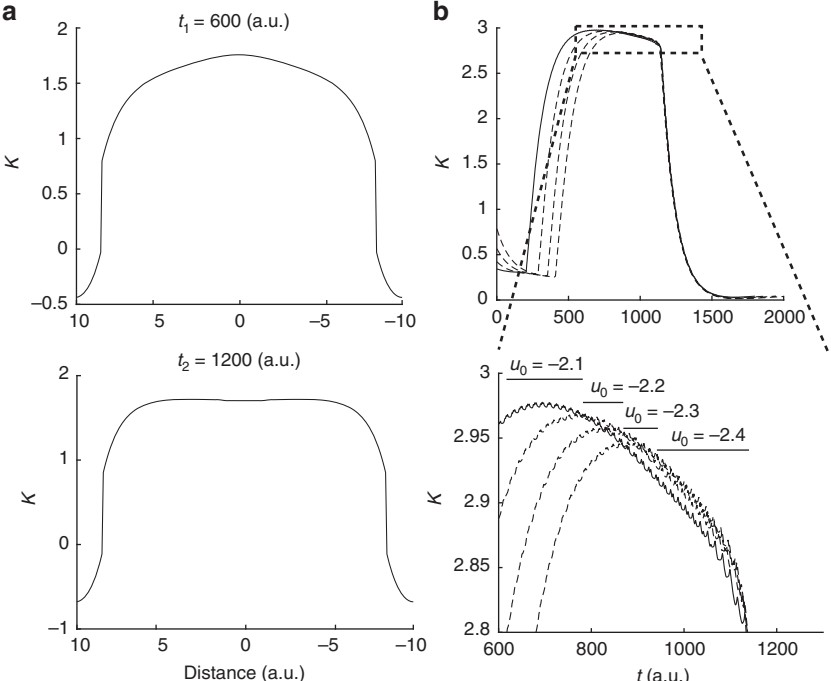

**Fig. 5** The spatial source of ictal spike-and-wave discharges. The variable $K$, which combines the slow permittivity variable and the average activity of the fast population, determines in part the spatial source of SWDs. The variable $K$ represents the instantaneous values of slowly evolving processes during the seizure and it can evolve differently for different parts of the field during a seizure, thus resulting in a moving source for the SWDs (Methods). **a** Evolution of the variable $K$ during a seizure indicated by two different snapshots ($t_1$ and $t_2$). **b** Top plot: the variable $K$ as a function of different excitability levels including $v_0 = -2.1$ (full line) and for values ranging from $-2.4$ to $-2.2$ (dashed lines), at a specific site of the Epileptor field model. Bottom plot: zoomed in view of the top plot. The maxima of the variable $K$ (black bars) computed over space, therefore corresponding to the sources of propagation, are obtained by comparing $K$ for different values of $u_0$ at each time point

same time. Indeed, it has been observed in ECoG data that the average seizure duration tends to be $2.3 \pm 0.4$ times longer than the duration of seizure propagation[18]. On the other hand, if the recruitment times are too long, a slowly propagating pulse appears instead, that is, the seizure offset slowly propagates across the Epileptor field model (Supplementary Fig. 6). In this case, one observes seizure termination as a slowly propagating event as mentioned earlier.

We also note that a few SWDs may still propagate throughout the Epileptor field model after the seizure termination. These SWDs propagate through the recruited brain areas (Fig. 2 right lower row), and are very similar to isolated propagating SWDs often observed in ECoG or SEEG recordings after seizure termination. The occurrence and the number of these SWDs after the seizure termination depends on the strength of the coupling function $\gamma_{12}$ in the temporal kernel $g$ (Supplementary Fig. 7). Nevertheless, we emphasize that seizure termination would still appear synchronous if one defines the time of seizure termination as the time of the last SWDs (Fig. 2).

**SWDs fail to propagate between regions with large delays.** As seen in Fig. 1, seizures do not always terminate simultaneously across the brain. Instead, many seizures evolve into the formation of ictal clusters. While seizure termination is still synchronous within each ictal clusters, long termination delays can occur across clusters. According to our model, long termination delays between different brain regions indicate a failure of the SWDs to trigger seizure termination synchronously in these regions. To reproduce the large delays in seizure recruitment and termination observed between ictal clusters (Fig. 1), we examined in more detail the role of heterogeneous connectivity between two

Epileptor field models (black arrow in Fig. 2 middle row). Such neural field architecture ("two-point connection") has been previously investigated in detail[36,37].

The Epileptor field model with a two-point connection architecture accounts for a seizure starting in a brain region, slowly propagating within this region, and recruiting another brain region with longer delays (Fig. 2). Synchronous termination can occur between separate Epileptor field models interacting via heterogeneous connections, provided that the heterogeneous connection strength $\gamma_{het,12}$ between the two fields is high enough. Since SWDs propagate independently in each field, pairwise correlation values just before seizure termination between LFP sites in two different fields are lower than for LFP sites in the same field (Fig. 6b). On the other hand, large termination delays can arise between separate Epileptor field models for lower heterogeneous connection strength $\gamma_{het,12}$ (Fig. 6b).

Next, we examine in more detail the role of the SWDs in triggering synchronous seizure termination. To quantify the ability of SWDs to propagate between sites in different fields at seizure termination, we computed the pairwise correlation values just before seizure termination between each spatial coordinate in the two connected Epileptor field models for different heterogeneous connection strength ($\gamma_{het,12}$) and excitability parameter ($u_{0,2}$) values of the second Epileptor field model (see Methods for details). We used an automated clustering method to identify clusters based on the pairwise termination delays only, i.e., without including the pairwise correlations (Fig. 6b). Other examples of simulated seizures and clustering results are shown in Supplementary Fig. 8. For each identified cluster, we computed the mean pairwise correlation and mean termination delay (Fig. 6c). The corresponding connection strength for the same clusters are shown in Fig. 6d. As expected, for large termination

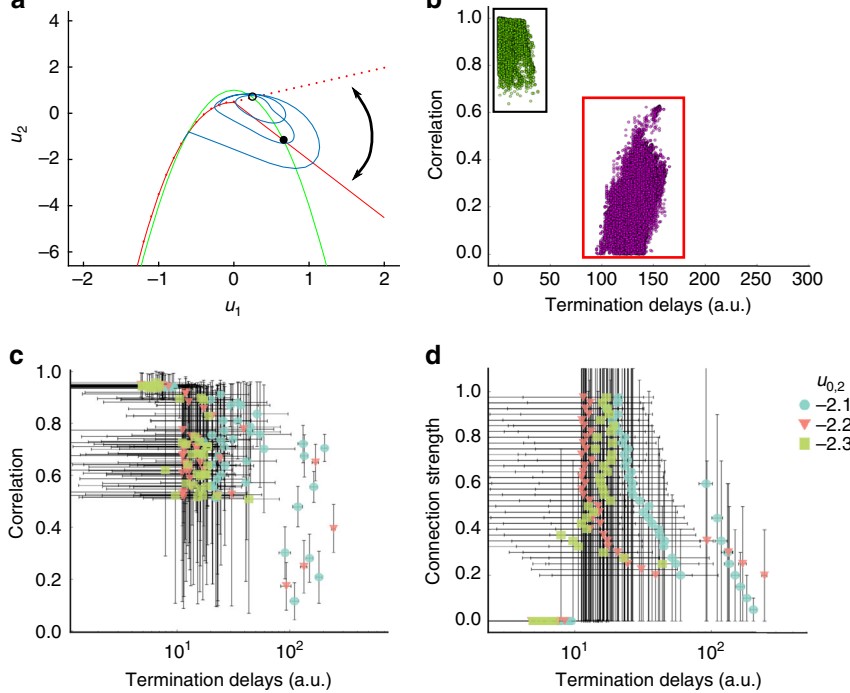

**Fig. 6** Synchronized seizure termination results from the fast propagation of ictal SWDs. **a** Phase space for the fast population ($u_1$ and $u_2$) of the Epileptor field model at seizure offset. The slope of the straight part of the red nullcline changes when a SWD occurs, transforming the stable fixed point (black dot) into an unstable fixed point (black circle) and allowing for fast oscillations to emerge (blue trajectory) and cross the separatrix via a homoclinic bifurcation. The homoclinic bifurcation results in the termination of the seizure. The curved arrow represents the change in the nullcline before and after a SWD. **b** Pairwise correlations between different sites in the neural field model as a function of seizure termination delays. Same seizure simulation as in Fig. 2, except that noise was added (see Methods). Black (red) rectangle: pairwise correlation for spatial locations in the same (different) Epileptor field model. Lower correlation values are obtained when the pairwise sites are in different fields (red rectangle). Green and violet dots indicate different clusters obtained via automated clustering (Methods) of the termination delays. **c** Mean pairwise correlation between different sites in the Epileptor field model as a function of termination delays for different values of the excitability ($u_{0,2}$) of the second field and of the heterogeneous coupling strength ($\gamma_{het,2}$). Each point corresponds to the mean for each one of the clusters obtained as in **b**. Each color/symbol indicates a different excitability value. The vertical and horizontal bars indicate the standard deviations in the corresponding coordinates. Note the log scale in the x-axis. **d** Same conventions as in **c**, except by showing instead mean pairwise connection strengths

delays, both the correlation values and the connection strengths are small. Small termination delays can lead to high correlation values when the two site pairs are in the same Epileptor neural field. These high correlations confirm the synchronization between SWDs in the same Epileptor field. Small termination delays can also be associated with lower correlation values when recruitment delays (i.e., seizure spread from one field to the other) are small and the site pairs are in different Epileptor field models. In addition, for small termination delays, some clusters can contain pairwise locations both from the same or from separate Epileptor field models (Fig. 6b). In these cases the average connection strength can be low as the homogeneous connection strength is ignored in the analysis. We note that long recruitment delays are obtained for only a small range of heterogeneous connection strengths in the Epileptor field model, and that most seizures have a small recruitment delay.

**Large termination delays correlate with weak connections**. We next checked our predictions for the role of SWDs and connection strength in seizure termination in SEEG data recordings. We analyzed 54 seizures from 13 patients (Methods). Six of these seizures contained large seizure termination time delays between different brain regions. Figure 7a shows the pairwise site correlation as a function of the termination delay for one seizure (same seizure shown in Fig. 1). Pairwise correlations were computed for electrode pairs belonging to different stereotactic electrodes in all

patients and seizures. As above, we numerically identified clusters based on the pairwise termination delays only, and we computed mean pairwise correlation values just before seizure termination and mean pairwise termination delays for each cluster. Other examples of human seizures and clustering results are shown in Supplementary Fig. 8. We found that large termination delays tended indeed to be associated with small correlation values (Fig. 7b), while small termination delays could lead to high and low correlation values.

As suggested by our simulations, we hypothesized that brain regions showing large termination delays are weakly connected via homogeneous and heterogeneous connectivity. To confirm our hypothesis, we processed structural and diffusion MRI data for these patients. We used the parcellation atlas and tractography methods to estimate the proximities (i.e., contacts being in contiguous brain regions, Supplementary Fig. 9) and number of fiber tracks between pairs of electrodes as a function of corresponding seizure termination delays (Methods). Using the same identified clusters, as in Fig. 7b, we found that large termination delays tended to be associated with contacts being in non-contiguous regions and with small number of tracks (Fig. 7c, d). We sorted the results for all patients into two groups consisting of small and large termination delays. We found significant differences for correlations, number of tracks, and proximity between the two groups (Mann–Whitney U-test for intra vs. inter clusters, $P < 0.01$ for both measures, Fig. 7e).

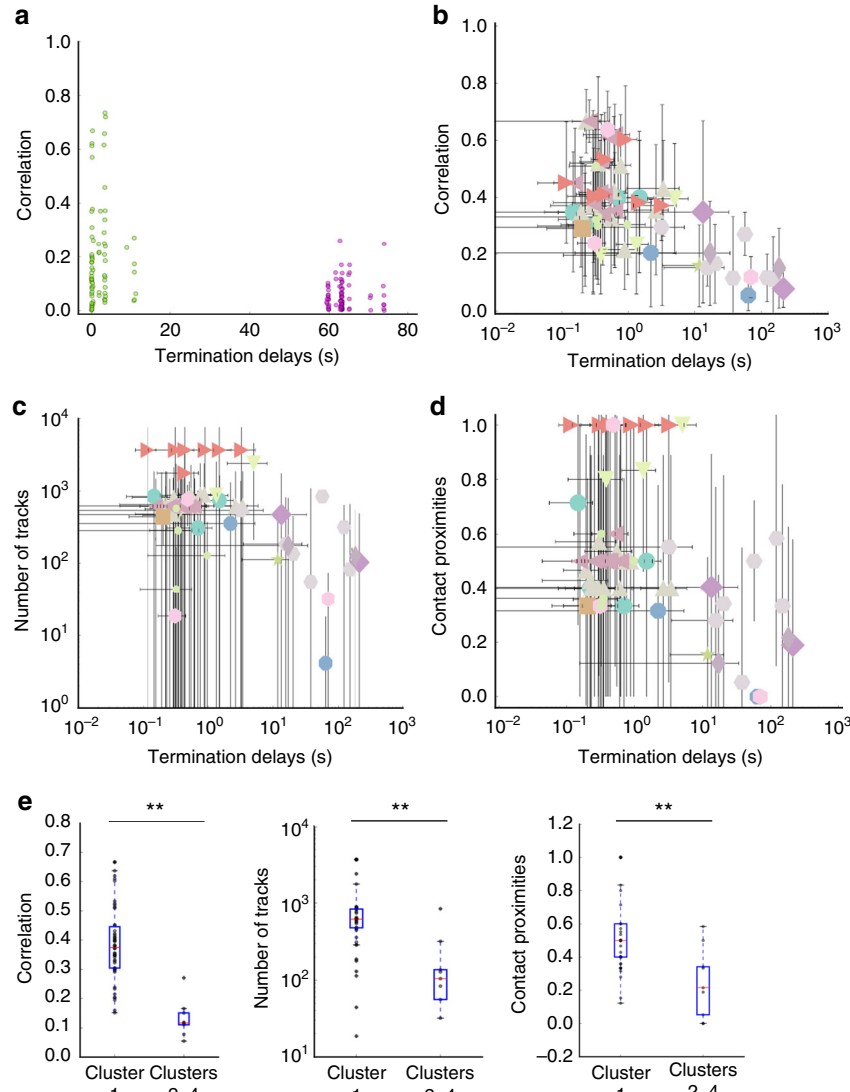

**Fig. 7** Large differences in seizure termination delays can arise between different ictal clusters involving distant and not well connected brain regions. **a** Pairwise correlations as a function of the pairwise termination delays for stereotactic contacts from an example seizure (Fig. 1). Green and violet dots identify clusters based on the pairwise termination delays. **b** Mean pairwise correlation between contacts as a function of the corresponding pairwise termination delays for all contacts, patients and seizures. Each point corresponds to the mean for each one of the clusters obtained as in **a**. Each color/ symbol indicates a different patient. The vertical and horizontal bars indicate the standard deviation in the corresponding coordinate. Note the log scale in the x-axis. **c** Mean pairwise number of tracks between contacts as a function of corresponding pairwise termination delays. Same conventions as in **b**. Note the log scale in both the x- and y-axes. **d** Mean pairwise contact proximities as a function of corresponding pairwise termination delays. A value of one indicates that all pairwise contacts in the cluster are in contiguous brain regions. Same conventions as in **b**. **e** Whisker plots for pairwise correlation, number of tracks, and contact proximities for cluster one (smallest termination delays) vs. the other clusters (longer termination delays) across patients. Statistically significant differences are indicated (**$P < 0.01$; Mann–Whitney U-test, $n = 60$)

To conclude, the ability of SWDs to propagate at seizure termination is correlated with synchronous seizure termination and high homogeneous or heterogeneous connection strength. This indicates that SWDs play a role in synchronous seizure termination. Conversely, low connection strength between two distant brain regions favors the occurrence of long termination delays, and thus the formation of ictal clusters and asynchronous seizure termination among them.

## Discussion
In this article we have introduced a new neural field model that unifies and explains the previously observed diversity in spatio-temporal dynamics of seizure initiation, propagation and termination and their multiple time scales. We have demonstrated how the interplay between temporal and spatial scales leads to: (i) slow propagation of an ictal wavefront whose speed is hampered by fast oscillations (LVFA), (ii) fast propagation of SWDs through coupled-oscillator dynamics, with either moving or spatially stationary sources, (iii) the formation of ictal clusters via stagewise recruitment with synchronous (within clusters) and asynchronous (across clusters) seizure termination. Furthermore, we confirmed in SEEG and tractography data recorded from patients with epilepsy the model's predictions that SWD propagation and connection strength correlate with the type of spatial patterns of seizure recruitment and termination. Our results shed new light on the dynamical mechanisms underlying the apparently contradictory diversity of spatiotemporal patterns across

different seizures and patients. Altogether, we propose a comprehensive model of spatiotemporal dynamics in epileptic focal seizures, contributing to more accurate individualized patient modeling and better therapeutic interventions.

Our earlier work on the Epileptor model introduced a taxonomy of seizure dynamics based on bifurcation normal forms for seizure initiation and termination[14], and more recently on the more general unfolding theory[38]. As expected, the development of an explicit taxonomy based on dynamical system theory for both spatial and temporal aspects of seizure dynamics is a very challenging problem. Tools and techniques well established in the finite dimensional world of dynamical systems (ODEs) do not trivially translate into the infinite dimensional evolution equations (PDEs). Nevertheless, our neural field extension of the original Epileptor model and the new findings stated above suggest that the spatiotemporal dynamics of seizure propagation and termination is restricted to a set of possible general cases. This issue is discussed in more detail in the Methods section (see also Supplementary Figs. 10–12).

Mixed connectivities in network architectures are common in large-scale brain models[39,40]. They were first introduced by Jirsa and Kelso[36], and have been used since then in the modeling of resting state networks[37] and evoked potentials induced by electrical stimulation[40]. In the neuroinformatics platform "The Virtual Brain"[41], such neural field architectures are implemented as surface-based modeling approaches, in which a high-resolution cortical surface is equipped with a neural field and homogeneous short-range and heterogeneous long-range connectivity. The latter connectivity is typically obtained from diffusion MRI-derived connectomes and may hold different biophysical mechanisms. In our previous work[3,42], we modeled the heterogeneous coupling between different brain areas via a slow permittivity coupling in different Epileptors. Here, long-range interactions through heterogeneous coupling were modeled via fast population activity ($u_1$) at different sites of the Epileptor field model. The two approaches are not incompatible, but rather complementary, as they account for different biophysical mechanisms: Fast coupling accounts for synaptic interactions, while slow permittivity coupling accounts for slowly evolving physiological processes such as extracellular ionic diffusion involving, e.g., potassium[26,43], its spatial buffering by glial cells[28,44], as well as increase, via long-range projections in remote regions, of its extracellular concentration via increased firing rate[28,34]. In terms of dynamics, fast coupling explains seizure dynamics such as slow seizure propagation, fast SWDs propagation, and synchronous seizure termination. On the other hand, slow permittivity coupling can reproduce very large delays of recruitment between distant brain regions, which are difficult to obtain in the model presented here (Fig. 6c, d). However, both models underline the importance of connection strength in seizure spatiotemporal dynamics. In particular, we explained here how brain regions showing low pairwise connection strength are, if recruited, less likely to show synchronous seizure termination and display low pairwise coherence, as experimentally observed by other groups, e.g., ref. [45]. We used here the number of fiber tracks as a proxy for connection strength. While the number of tracks may not reflect the connection strength between two brain regions[46], the tractography methods we used have been shown to robustly match the total cross-sectional area of the white matter fiber bundles between brain regions[47]. The actual relationship between connection strength and number of tracks, while likely monotonic, may be nonlinear, e.g., logarithmic as suggested by Fig. 7c.

As stated earlier, recent studies have provided contrasting evidence regarding the emitting source and direction of SWDs. In one study[12], it has been argued that ictal SWDs are emitted by a moving source, which consists of the slowly propagating ictal wavefront. In contrast, another study[8] has provided evidence for SWDs being emitted by a spatially stationary source consisting of the identified seizure onset area in each seizure. Our results and predictions show that both scenarios (moving and stationary sources) are possible. A critical feature allowing this possibility is the emergence of coupled-oscillator dynamics. Once different sites in the Epileptor field are recruited into the ictal state and transition into oscillatory activity, a network of coupled oscillators is formed. Changes in the location of the source emitting SWDs can result from reorganization of the phases of the neural oscillators. This is a crucial difference between our model and the two previously mentioned studies. In both refs. [12] and [8], the SWDs are only emitted by a small brain region, which can be moving or not (Supplementary Figs. 10 and 11). If, during a seizure, the ictal activity in this brain region terminates, for example by electrical stimulation, the seizure will immediately end across the brain. In our model, however, the termination of a seizure might require that ictal activity is abated in all or a much larger set of recruited brain areas (Supplementary Fig. 12). These points are discussed in more detail in the Methods section. This prediction can be verified using electrical stimulation for seizure abatement in localized areas, and is critical for the development of closed-loop systems for seizure control[6]. In particular, our model demonstrates the importance of early seizure detection (e.g., ref. [48]), as this is one of the most favorable times to abate seizures as the recruited area is still small and easier to stimulate. Finally, the emergence of these coupled-oscillator dynamics is to be contrasted with pure traveling wave dynamics in excitable media[23] underlying the slow propagating ictal wavefront. While the propagation dynamics of SWDs can be more complex, seizures tend to propagate as wavefronts in excitable media, in agreement with experimental observations that seizures tend to have a common spatial origin in the same patient.

As a phenomenological model, the Epileptor field model identifies the invariant features that constrain the observed dynamics and may inform the development of detailed biophysical models. It has been suggested that the fast population comprises excitatory neurons, while the population on the intermediate time scales mostly comprises inhibitory neurons[14,49]. Here, we suggest another possibility: Small networks of coupled excitatory and inhibitory neurons could be represented by the fast population dynamics, without particularly assigning one variable to a single excitatory or inhibitory population. Instead, the feedback-loop circuitry in which excitatory and inhibitory neurons are embedded allows for the oscillatory behavior captured by the fast population in the Epileptor model. The transition to the SWD activity could be triggered by various factors including increase of extracellular potassium[28,34], which can impair inhibitory neuron activity once depolarization block sets in. In the Epileptor field model, the SWD activity increases because of the fast population activity ($u_1$) through a slow temporal convolution ($g$), which increases and then reaches a plateau during the seizure. These dynamics affecting SWDs could be instantiated by, for example, changes in extracellular potassium concentration. The SWDs produced by the second population would emerge from the contribution of excitatory neurons only. For the coupling functions in the Epileptor field model, the coupling kernels all introduce positive terms on the different populations of the Epileptor field models (i.e., an excitatory effect), while the inhibitory terms are locally embedded in the Epileptor equations. While we used three different coupling functions, that does not mean they would necessarily be conveyed by different populations of excitatory neurons in a detailed biophysical implementation. Rather, transitioning between the different steps of the seizure would in turn recruit theses neurons into different dynamical regimes, and result in different

dynamical activity being propagated through the same axons. We also note that here we have focused on epileptic seizures that show SWDs. We hope to address in the future the case of seizures that do not include ictal SWDs, but remain instead within the dynamics of low-voltage fast-activity oscillations[19,28,50,51].

Patient-specific structural heterogeneous connectivity has recently been used in brain network modeling aiming at personalized medicine, modeling and, in particular, seizure recruitment[1,3,52]. Other methods have focused on the neural field formulation[8,11] to capture SWD propagation. However, integrating both types of connectivity in a single model can lead to qualitatively different spatiotemporal dynamics than suggested by the combination of dynamics predicted by models using either homogeneous or heterogeneous connectivity[53]. By including both homogeneous and heterogeneous connectivity in the same model, we were able to reproduce the diversity of seizure initiation, propagation and termination patterns, and we have demonstrated how different time scales indeed interact together in seizures dynamics. The disadvantage of integrating both types of connectivity is the computational cost of large-scale simulations. However, with the increasing computational resources and the advance of neuroinformatic methods such as The Virtual Brain[41], such modeling tools will be accessible to assist clinicians' diagnosis in their daily practice. By improving the accuracy of neural dynamics models across different temporal and spatial scales, we hope to improve the virtualization potential of epileptic brains, the success rate of therapeutic interventions in patients with pharmacologically resistant focal epilepsies, and contribute to the development of closed-loop neuromodulation systems for seizure control.

## Methods

**Patient selection and data acquisition.** A total of 13 drug-resistant patients (6 men, mean age 32.8 years, range 22–63 years) with different types of partial epilepsy were selected (Supplementary Table 1). SEEG electrodes were implanted in the regions suspected to be in the epileptogenic zone. Each electrode had 10–15 contacts (length: 2 mm, diameter: 0.8 mm, contacts separation: 1.5 mm). SEEG signals were recorded with a 128-channel Deltamed™ system (sampling rate: 256 Hz, hardware bandpass filter: 0.16–97 Hz). To determine electrode positions, an MRI was performed after electrodes implantation (T1 weighted anatomical images, MPRAGE sequence, TR = 1900 ms, TE = 2.19 ms, $1.0 \times 1.0 \times 1.0$ mm$^3$, 208 slices) using a Siemens Magnetom Verio 3T MR-scanner. To reconstruct patient specific connectomes (DTI-MR sequence, angular gradient set of 64 directions, TR = 10.7 s, TE = 95 ms, $2.0 \times 2.0 \times 2.0$ mm$^3$, 70 slices, b weighting of 1000 s/mm$^2$, diffusion MRI images were also obtained on the same scanner. The study was approved by the Comité de Protection (CPP) Marseille 2, and all patients signed an informed consent form.

**Data processing.** To quantify the proximity and number of tracks between electrodes, structural and diffusion MRI data were obtained via a processing pipeline to derive individualized cortical surface and large-scale connectivity[53]. Cortical and subcortical surfaces were reconstructed along with volumetric parcellations using the Desikan–Killiany atlas, with the cortical regions subdivided in four[54] (280 cortical regions and 17 subcortical regions). We obtained electrode positions by coregistering the parcellation with the MRI scan, and assigning each contact to the region containing most of the reconstructed contact volume. Using a binary proximity matrix that indicates if two regions are contiguous in the brain atlas, we then computed the binary proximity between the corresponding pairwise electrodes (Supplementary Figure 9). To compute the number of tracks between electrodes, head-motions and eddy-currents were corrected in diffusion data. Fiber orientation was estimated with constrained spherical deconvolution, and $2.5 \times 10^6$ streamlines were obtained by probabilistic tractography. We used the anatomically-constrained tractography (ACT) and the spherical-deconvolution informed filtering of tractograms (SIFT) frameworks to improve reproducibility and biological accuracy[47]. The number of tracks between two pairwise electrodes was then obtained by summing the number of tracks whose extremities belong to the corresponding pairwise brain regions.

**Data analysis.** Spike-and-wave events were extracted from raw SEEG signals by band-pass filtering between 1 and 10 Hz (fifth-order forward-backward Butterworth filter). For each pair of contacts, the two-second window before the earliest offset of the two contacts were selected for both contacts (we obtained similar

results by using two-second windows at 2, 4, 6, 8, and 10 s before the earliest seizure offset) and correlation computed. Proximities and number of tracks between pairwise contacts were obtained between associated regions from the parcellation (see Data processing). Both contacts were allowed to be in the same region, as self-connection of the binary and connectivity matrices were not null, but they had to belong to the different stereotactic electrodes to better span the spatial dimension of the cortex. In addition, we only took into consideration recording sites that were either in the gray cortical matter or in the hippocampus/amygdala. We used mean-shift clustering to group pairwise delays and obtain mean and standard deviation of each group for the different measures.

**Epileptor neural field model.** We extended spatially a five-dimensional model able to reproduce the LFP dynamics of epileptic seizures, known as the Epileptor model[14,55]. The model comprises three different time scales interacting together. The slowest time scale is responsible for leading the autonomous switch between interictal and ictal states, and is driven by a slow permittivity variable. The fastest and intermediate time scales are two coupled oscillators accounting respectively for the LVFA oscillations and SWDs. We introduce here the integral neural field form of the Epileptor model, to account for propagation through short-range connectivity. The couplings are established to satisfy minimal constraints imposed by physiology and lowest order approximations of local connectivity kernels ([37]; see also General methodological considerations below). The five dimensions of the original model are now described as a five-dimensional neural field representing the fast activity ($u_1(x,t)$, $u_2(x,t)$), the spike-and-wave activity ($q_1(x,t)$, $q_2(x,t)$), and the slow permittivity variable $v(x,t)$ at position $x$ and time $t$ as follows ($x$ and $t$ are omitted for simplicity)

$$\partial_t u_1 = u_2 - f_1(u_1, q_1) - v + I_1 + \gamma_{11} w_1 * S(u_1, \theta_{11}) \\ + \sum_j \gamma_{\text{het},j} w_{\text{het}} S(u_{1,j}, \theta_{\text{het}}) \tag{1}$$

$$\partial_t u_2 = 1 - 5u_1^2 - u_2 \tag{2}$$

$$\partial_t v = \frac{1}{\tau_0}(4(u_1 - u_0(x)) - v) \tag{3}$$

$$\partial_t q_1 = -q_2 + q_1 - q_1^3 + I_2 + 0.002g(u_1) \\ -0.3(v - 3.5) + \gamma_{22} w_2 * S(q_1, \theta_{22}) \tag{4}$$

$$\partial_t q_2 = \frac{1}{\tau_2}(-q_2 + f_2(q_1)), \tag{5}$$

with

$$g(u_1) = \int_{t_0}^t e^{-(t-s)/\tau_{12}}(a_{12}u_1 + \gamma_{12}w_{12} * S(u_1, \theta_{12}))\text{d}s \tag{6}$$

$$f_1(u_1, q_1) = \begin{cases} u_1^3 - 3u_1^2 & \text{if } u_1 < 0 \\ (q_1 - 0.6(v-4)^2)u_1 & \text{if } u_1 \geq 0 \end{cases}$$

$$f_2(q_1) = \begin{cases} 0 & \text{if } q_1 < -0.25 \\ 6(q_1 + 0.25) & \text{if } q_1 \geq -0.25, \end{cases}$$

where $\partial_t$ denotes the partial derivative with respect to time, and $I_1 = 3.1$, $I_2 = 0.45$, $\tau_0 = 2857$, $\tau_2 = 10$, $\tau_{12} = 1/0.01$, $\theta_{11} = -1$, $\theta_{22} = -0.5$, $\theta_{12} = -1$, $\theta_{\text{het}} = -1$, $a_{12} = 3$. $\tau_{12}$ represents the speed of the temporal integration in Eq. 6. We note that in the original Epileptor $a_{12} = 1$, the main difference being that SWD propagation is shifted toward the beginning of the seizure as $g$ increases more rapidly (Supplementary Fig. 13). The parameter $u_0(x)$ represents the excitability of the Epileptor at position $x$. Different values of $u_0(x)$ are used throughout this article. If $u_0 > -2.91$, the Epileptor is epileptogenic and able to trigger seizures autonomously. Otherwise, the Epileptor stays in a healthy equilibrium state.

Two local coupling functions $\gamma_{11} w_1 * S(u_1, \theta_{11})$ and $\gamma_{22} w_2 * S(q_1, \theta_{22})$ are added between variables $u_1$ and $q_1$ respectively. An additional coupling function $\gamma_{12} w_{12} * S(u_1, \theta_{12})$ spatially extends the action of $u_1$ in the temporal convolution $g$. The coupling term $S(u, \theta)$ is interpreted as a firing rate function, and is chosen here to be the Heaviside coupling function $S(u, \theta) := H(u - \theta)$. The spatial convolution $*$ is defined as

$$w * S(u, \theta) := \int_{-\infty}^{+\infty} w(y)S(u(x-y,t), \theta)\text{d}y,$$

with $w(y)$ representing the connection strength between sites separated by a distance $y$ in a given Epileptor field. $w(y)$ is chosen to be isotropic, integrable and homogeneous. Here, we used a Laplacian local connectivity kernel $w(x) := w_1 = w_2 = w_{12} = e^{-|x|}/2$. The explicit form of the coupling functions is based on statistics of synaptic density distribution (see also General methodological

considerations below). Such choice is common in the neural field literature, and solutions of such systems have been shown to typically yield traveling front and wave dynamics. Unless specified otherwise, $\gamma_{11} = \gamma_{22} = \gamma_{12}/10 = 1$.

Finally, heterogeneous connectivity between the local and distant Epileptor field models, indexed by $j$, is captured by the term $\sum_j \gamma_{\text{het},j} w_{\text{het}} S(u_{1,j}, \theta_{\text{het}})$, with $u_{1,j}$ corresponding to the $j$th Epileptor field. The heterogeneous connectivity kernel is chosen Gaussian: $w_{\text{het}} := 1/\sqrt{8\pi} e^{-x^2/8}$. Unless specified otherwise, $\gamma_{\text{het},j} = 0.3$ for all $j$. Parameter space explorations of the effects of these parameters on the dynamics of the Epileptor field model are shown in Figs. 3, 6, and Supplementary Fig. 1, and were found to be robust over a large range of values.

**Numerical implementation.** Direct numerical simulations of the Epileptor field model used a 4th-order Runge-Kutta integration scheme with fast Fourier transform for the calculation of convolutions[56]. The Epileptor field model activity was set to the fixed point, and a stimulation pulse of strength 1 and spatial width 1.57 was added to $I_1$ at time $t = 400$ for a duration $\Delta t = 10$. The field size was set to $6\pi$ (a.u.). For Fig. 6, a zero mean white Gaussian noise process with a variance of 0.001 was added to the variables $q_1$ and $q_2$ to better reproduce the distribution of correlation values.

**Seizure propagation speed for the fast population.** To evaluate the speed of propagation of the seizure as function of parameters $\theta_{11}$, $u_0$, and the variable $v$, we analyzed the jump-up solution from the interictal state to the ictal state. We therefore focused on the part of the phase space where $u_1 < 0$ (i.e., $f(u_1, q_1) = u_1^3 - 3u_1^2$), following ref. [42]. The system 1–6 becomes

$$\partial_t u_1 = u_2 - u_1^3 + 3u_1^2 - v + I_1 + \gamma_{11} w_1 * S(u_1, \theta_1 1) \tag{7}$$

$$\partial_t u_2 = 1 - 5u_1^2 - u_2 \tag{8}$$

$$\partial_t v = \frac{1}{\tau_0}(4(u_1 - u_0) - v). \tag{9}$$

Taking the Fourier transform, $\mathcal{F}\{w\}(k) := \int_{\mathbb{R}} e^{ikx} w(x) dx$, of Eq. 7 gives[20]

$$ik\mathcal{F}\{u_1\}(k) = \mathcal{F}\{u_2 - u_1^3 + 3u_1^2 - v + I_1\}(k) \\ + \gamma_{11} \mathcal{F}\{w_1\}(k)\mathcal{F}\{S(u_1, \theta_{11})\}(k)$$

according to the Fourier transform of a convolution.

For our choice of connectivity kernel $w(x) = e^{-|x|}/2$, $\mathcal{F}\{w_1\}(k) = (1 + k^2)^{-1}$, which gives

$$\gamma_{11}\mathcal{F}\{S(u_1, \theta_{11})\}(k) = (1 + k^2)(ik\mathcal{F}\{u_1\}(k) \\ + \mathcal{F}\{u_1^3 - 3u_1^2 - u_1 - I_1 + v\}(k)).$$

By inverse Fourier transform we obtain

$$\gamma_{11} S(u_1, \theta) = (1 - \partial_{xx})(\partial_t u_1 + u_1^3 - 3u_1^2 - u_2 - I_1 + v). \tag{10}$$

We introduced the change of variable $\xi = x + c_1 t$, and defined the traveling front solution by $U_1(\xi) = u_1(x + c_1 t) = u_1(x,t)$, and similarly for $u_2$ and $v$. By the chain rule,

$$\partial_t u_1(x + c_1 t) = \frac{d\xi}{dt}\frac{dU_1(\xi)}{d\xi} = c_1 U_1',$$

$$\partial_{xx} u_1(x + c_1 t) = \frac{d\xi}{dx}\frac{d}{d\xi}\left(\frac{d\xi}{dx}\frac{dU_1(\xi)}{d\xi}\right) = U_1'',$$

and similarly for $u_2$ and $v$. By differentiation rules, Eqs. 8, 9, and 10 become

$$\gamma_{11} S(U_1, \theta_{11}) = U_1 U_1''(-3U_1 + 6) + U_1'(-6U_1'(U_1 - 1) + c_1) \\ -c_1 U_1''' + U_1^3 - 3U_1^2 - I_1 - U_2 + U_2'' + V - V''$$
$$c_1 U_2' = 1 - 5U_1^2 - U_2$$
$$c_1 V' = \frac{1}{\tau_0}(4(U_1 - u_0) - V).$$

Using averaging methods, we focused on the fast time scale by setting $V' = 0$. Thus $V = \overline{V}$ is a constant parameter. Additionally we recast the system as the

following system of ODEs

$$U_1' = E_1$$
$$E_1' = H_1$$
$$H_1' = -\frac{1}{c_1}\left(U_1 H_1(3U_1 - 6) + E_1(6E_1(U_1 - 1) - c_1 + \frac{10U_1}{c_1}\right.$$
$$\left. + U_2 + \frac{E_2}{c_1} - U_1^3 + 3U_1^2 + I_1 - \overline{V} + \gamma_{11} S(U_1, \theta_{11})\right)$$
$$U_2' = E_2$$
$$E_2' = \frac{1}{c_1}(-10E_1 U_1 - E_2). \tag{11}$$

Using this system of equations, we constructed the front numerically by a shooting method. We looked for a heteroclinic solution that links the interictal state fixed point (black full point in Supplementary Fig. 3) to the stable limit cycle (full black cycle in Supplementary Fig. 3). Suppose the speed of the front is given by $c_1^*$; we numerically integrated the system 11 for a given speed $c_1$. Depending on whether $c_1 < c_1^*$ or $c_1 > c_1^*$, we obtained the blue or the green trajectory, respectively, in Supplementary Fig. 3. We now have an optimization problem where we are looking for the speed $c_1^*$ that minimizes the distance between the stable limit cycle and the simulated trajectory, that is the speed $c_1^*$ for which we obtain a front solution. This optimization problem was solved numerically by the Nelder-Mead optimization method, and the front solutions were integrated using an Euler integration scheme.

**Seizure propagation speed for the reduced fast population.** We demonstrate that the fast oscillatory activity at the beginning of the seizure reduces propagation speed by deriving the speed of propagation of the seizure when this activity is neglected. We first reduced the Epileptor field model (Eq. 7) to two degrees of freedom by setting $\dot{u}_2 = 0$, thereby replacing the ictal oscillatory activity by its average[42]

$$\partial_t u_1 = -u_1^3 - 2u_1^2 + 1 + I_1 - v \\ + \gamma_{11} w_1 * S(u_1, \theta_{11})$$
$$\partial_t v = \frac{1}{\tau_0}(4(u_1 - u_0) - v).$$

Proceeding as above, we obtained the following system of ODEs

$$U_1' = E_1$$
$$E_1' = H_1$$
$$H_1' = -\frac{1}{c_1}\left(U_1 H_1(3U_1 + 4) + E_1(E_1(6U_1 + 4) - c_1) \right.$$
$$\left. - U_1^3 - 2U_1^2 + 4.1 - \overline{V} + \gamma_{11} S(U_1, \theta_{11})\right).$$

We then used the same shooting method as above to obtain Supplementary Fig. 5.

**SWD propagation speed for the second population.** To derive the speed of the SWDs in population 2, we first need to isolate population 2 from the full Epileptor field model. To do so, we first recast the term $0.002g(u_1)$ in Eq. 4 as $0.002g(u_1) := 2q_3$. We therefore have

$$q_3 = 0.001 \int_{t_0}^{t} e^{-(t-s)/\tau_{12}}(3u_1 + \gamma_{12} w_{12} * S(u_1, \theta_{12})) ds.$$

By property of the Laplace transform, $\mathcal{L}\{f\}(s) := \int_0^\infty e^{-st} f(t) dt$, of a convolution,

$$\mathcal{L}\{q_3\}(s) = \frac{0.001}{s + 1/\tau_{12}}\mathcal{L}\{3u_1 + \gamma_{12} w_{12} * S(u_1, \theta_{12})\}(s).$$

Back to the time domain,

$$\partial_t q_3 + \frac{q_3}{\tau_{12}} = 0.001(3u_1 + \gamma_{12} w_{12} * S(u_1, \theta_{12})),$$

giving

$$\partial_t q_3 = -\frac{q_3}{\tau_{12}} + 0.001(3u_1 + \gamma_{12} w_{12} * S(u_1, \theta_{12})).$$

We performed an approximation using averaging methods along the left branch of the nullcline $\dot{q}_1 = 0$ (red line in Fig. 4b), to separate the time scales in

Eqs. 4 and 5 by setting $\dot{v} = 0$ and $\dot{q}_3 = 0$, giving $v = \overline{v}$ and $q_3 = \overline{q}_3$

$$\partial_t q_1 = -q_2 + q_1 - q_1^3 + I_2 + K(\overline{q}_3, \overline{v}) + \gamma_{22} w_2 * S(q_1, \theta_2) \quad (12)$$

$$\partial_t q_2 = \frac{1}{\tau_2}(-q_2 + f_2(q_1)), \quad (13)$$

with $K(\overline{q}_3, \overline{v}) = 2\overline{q}_3 - 0.3(\overline{v} - 3.5)$ a constant.

For the spatial part of the Epileptor field model which is in the ictal state, the values of $\overline{q}_3$ and $\overline{v}$ are such that the recruited Epileptors act as a chain of coupled oscillators (see Fig. 2). In the discretized problem, for contiguous coordinates $x_i$ and $x_j$ of the field, and two corresponding oscillators $i$ and $j$ with initial conditions $(q_1^i, q_2^i)$ and $(q_1^j, q_2^j)$, the speed of propagation of the SWD between the two coordinates is approximated by

$$c_2 := \frac{\Delta x}{\Delta T(q_1^i, q_1^j)} = \frac{x_j - x_i}{T(q_1^j) - T(q_1^i)}, \quad (14)$$

where $T(q_1^i)$ is the time for oscillator $i$ of the discretized problem to go from fixed point $q_1^i$ to the knee of the left branch of the cubic nullcline $q_1^c$ (red nullcline of Fig. 4b; note how the blue trajectory follows the left branch of the red nullcline). The coordinates $x_j$ and $x_i$ are chosen as in the discretized problem used for the numerical simulations of the full model.

To calculate the speed of propagation of SWDs $c_2$, we need to evaluate $T(q_1^i)$ in Eq. 14. To do so, we neglect the effect of coupling on the speed of propagation of the SWDs (i.e., $\gamma_{22} = 0$). This is confirmed by simulation (see Fig. 4c, d). On the left branch of the cubic nullcline, we have $q_1 < -0.25$, therefore $f_2(q_1) = 0$. In addition, since the trajectory follows closely the cubic nullcline, we set $\partial_t q_1 = 0$. Eqs. 12 and 13 at coordinate $x_i$ give

$$q_2 = q_1 - q_1^3 + I_2 + K(\overline{q}_3, \overline{v}) + \gamma_{22} w_2 * S(q_1, \theta_2) \quad (15)$$

$$\frac{1}{dt} = \frac{1}{\tau_2} \frac{-q_2}{dq_2}. \quad (16)$$

From Eq. 15, $dq_2 = dq_1(1 - 3q_1^2)$, and by substitution in Eq. 16, we obtain

$$dt = \frac{\tau_2(1 - 3q_1^2)}{-q_1 + q_1^3 - I_2 - K(\overline{q}_3, \overline{v})} dq_1.$$

Integrating between $q_1^i$ and $q_1^c$,

$$T(q_1^i) = \int_{q_i}^{q_c} \frac{\tau_2(1 - 3q_1^2)}{-q_1 + q_1^3 - I_2 - K(\overline{q}_3, \overline{v})} dq_1. \quad (17)$$

The speed of propagation is then evaluated using Eq. 14 for the two parameters investigated in the study (instantaneous phase, and $K$) by integrating numerically Eq. 17 for different values of $K$ with fixed initial phase $q_1^i$ (Fig. 4c), and for different values of the initial phase $\phi = q_1^i$ with fixed $K$ (Fig. 4d).

We note that the above approximation along the left branch of the cubic nullcline worsens when $K$ increases. Indeed, if $K$ increases, the unique fixed point of the system with the approximation $f_2(q_1) = 0$, for all $q_1$, which is given by the crossing of the flat green nullcline ($\dot{q}_2 = q_2 = 0$) and the right branch of the red cubic nullcline ($\dot{q}_1 = 0$), is shifted downwards. This moves the blue trajectory away from the nullcline as $K$ increases (Fig. 4b, inset), worsening the approximation.

**General methodological considerations**. The development of an explicit taxonomy based on dynamical system theory for both spatial and temporal aspects of seizure dynamics is a challenging problem. Canonical equations and normal forms exist for ODEs, and normal form theory is well developed using a variety of approaches. For the original Epileptor neural mass model, the authors followed initially the conventions used by ref. [57], and more recently used the more general normal form and unfolding theory[38]. Normal forms are of importance in the qualitative theory of differential equations. Using the theory of singularities of differentiable mappings, one can determine which number of terms of the normal form is sufficient to describe the bifurcation of stationary points and periodic solutions up to topological equivalence. Typically, the more generic the analysis is, the fewer the captured data features. This point is evident from ref. [38], where the authors used normal form theory to unfold the (Co-dimension 3) singularity of the Bogdanov–Takens point and identify the topology of qualitative behaviors in its neighborhood, but focused only on the first Epileptor population (seizure onset, offset and slow-variable dynamics).

The extension of canonical approaches to the spatial domain using neural field equations is mathematically significantly less developed. Tools and techniques, well established in the finite dimensional world of dynamical systems (ODEs), do not trivially extend to infinite dimensional evolution equations (PDEs). There are some notable exceptions such as the Birkhoff normal form of Hamiltonian systems,

allowing us to view the nonlinear evolution equations in the neighborhood of a stationary solution as small perturbations of infinite dimensional integrable ODEs. Generally, however, when dealing with spatiotemporal evolution equations, these are constrained to providing systematic expansions of the spatial and temporal operators, as well as the polynomial terms. These then lead to basic or fundamental equations, for which however their canonical nature in the sense of normal form theory is not rigorously justified. Examples include fundamental equations that have been investigated for decades including the linear Ginzburg–Landau equations, as well as the famous nonlinear Schrödinger equation, Burgers equation, Korteweg-de Vries equation and several others. These equations are fundamental, but they are not canonical in the sense of normal form theory in ODEs.

A critical aspect for the discussion of our neural field extension of the Epileptor neural mass model is the introduction and treatment of short and long-range spatial connectivity kernels. Specifically, there are two fiber systems to be considered: intra- and cortico-cortical fibers[37]. Most of the neural field literature is concerned with intracortical connectivity, which is captured by translationally invariant integral kernels (i.e., homogeneous connectivity). To be integrable, the kernels must decay fast enough over space, and are often represented as exponentially decaying functions[37]. These choices are not arbitrary, but informed by statistical analyses of detailed anatomical data from lateral short-range connectivity in the mouse[21] and their extrapolation to the human[22].

Beyond integrability, approximations based on expansions can further simplify the specification of spatial kernels. Here, when assuming an exponential spatial kernel (i.e., $\sim \exp(-|x|)$) for intracortical connectivity, we focus on local first-order and second-order expansions as a first approximation. The relationship between the expansion terms and the integral kernel for exponential kernels is captured by the equations (2.9) and (2.11) in ref. [37]. Solutions of neural field models based on these types of kernels have been shown to typically yield traveling front and wave dynamics[20]. Excitatory on / inhibitory off (e.g., "Mexican hat") organization that results from expansions of connectivity kernels of higher order than two are thus not included in our Epileptor neural field extension. Nevertheless, inhibitory effects are captured in our model by local connectivity already present in the original Epileptor ensemble equations. Our treatment is thus systematic in the sense that there is an approximation of connectivity with truncation criteria and an error term, where the latter converges to zero as the order of the expansion increases. Nevertheless, there are choices to be made regarding the coupling of the state variables. As stated above, these choices can be informed by anatomical/physiological data (e.g., ref. [21]) and considerations of local dynamics.

The second fiber system comprises the long-range corticocortical fibers and is translationally variant in space (heterogeneous connectivity). This system cannot be approximated locally by definition and needs to be tackled by empirically informed approaches[36]. This has been demonstrated for the first time in Jirsa et al.[58] and has become the basis for connectome-based brain network modeling. A characteristic of large-scale brain networks is the importance of time delays via signal transmission, that have been shown to alter the stability of the spatiotemporal patterns, but not the spatial patterns themselves[59–62]. Focusing here on the spatial pattern formation, we ignore the time delays, but incorporate the long-range corticocortical connectivity in our Epileptor neural field extension. Our choice of connectivity was informed by the known long-range fiber systems estimated from tractography.

In summary, we treat connectivity and its properties as systematically as possible given the current state of knowledge in pattern-forming systems and brain anatomy/physiology. Our treatment is not canonical in the rigorous sense of previous treatments of neural mass models[38], but currently it represents in our opinion the best compromise of generic and anatomically/physiologically informed approaches.

Nevertheless, our neural field extension of the original Epileptor model and the resulting analysis also suggests that the spatiotemporal dynamics of seizure propagation and termination is restricted to a set of possible general cases. We elaborate in more detail on two main related aspects:

(A) How can seizures often terminate synchronously across large brain regions given that their propagation across these regions is orders of magnitude slower? As demonstrated in Jirsa et al.[14], seizures can be characterized in dynamical terms by planar bursters, with only four possible onset bifurcations (saddle-node on invariant circle, saddle-node, supercritical and subcritical Hopf) and four possible offset bifurcations (saddle-node on invariant circle (SNIC), supercritical Hopf, fold limit cycle, and saddle homoclinic). We examine here the spatial aspects of the above onset and offset bifurcations and their implications to seizure propagation and termination.

Three of the onset bifurcations (SNIC, supercritical and subcritical Hopf) do not occur with a baseline shift, but with an oscillation around an equilibrium point. For these three bifurcations, we have two scenarios for seizure propagation after onset: either the oscillation is sustained for more than one cycle (Supplementary Fig. 10), or only a single cycle of excitation occurs (Supplementary Fig. 11a). The sustained oscillation scenario has to be discarded. That is because in this case one would see slowly propagating ictal waves emanating from the seizure onset area that would recur many times during a single seizure (Supplementary Fig. 10), something that, as far as we know, is not commonly seen in human seizures or animal models (typically, there is only a single ictal wavefront propagation event within a seizure). The second scenario (a single oscillation cycle) corresponds to a pulse propagation, illustrated in Supplementary Fig. 11a. For a propagating pulse,

the value of the variable of the field responsible for the pulse propagation is the same ahead of the front (non-recruited regions) and behind it (recruited regions). Hence, the activity in the recruited regions is driven by the propagating pulse, i.e., the ictal activity generated by recruited regions is directly dependent of the input activity originating from the propagating pulse (even though interactions between the pulse and the recruited surrounding can contribute to the oscillatory features and dynamics of the seizure). The two immediate consequences of such scenario are: (i) if the propagating pulse dies out, the seizure will immediately stop everywhere in the field (synchronous termination), and (ii) the source location of the spike-and-wave discharges (SWDs) is the pulse location. There is an important exception to consequence (i), which occurs if the field exhibits hysteresis. We come back to both points (i) and (ii) in the next section. For completeness, we note that it is possible to have a saddle-node onset bifurcation with front propagation immediately followed by an offset bifurcation, but it leads back to the scenario of pulse propagation we just discussed (Supplementary Fig. 11d).

The last possible onset bifurcation (saddle-node) allows for front propagation (Fig. 3a, Supplementary Fig. 12a). After the saddle-node bifurcation occurs at a given site, activity in this site jumps to a high level, which is then propagated as a front to the surrounding field via local projections instantiated by the local coupling function (e.g., $S(u_1,\theta_{11})$ in the Epileptor field model, Eq. 1). In this front propagation case, there are two possible ways to obtain a synchronous termination: (i) a global instability of the front (Supplementary Fig. 12b); This requires a global change of field parameters. We consider this solution unlikely because of the difficulty of identifying such a widespread global mechanism. For example, subcortical structures that project diffusively to many areas could play this global role. However, there are many examples of seizures in cortex that terminate synchronously, even though subcortical structures do not appear to have been recruited. (ii) One of the four possible local offset bifurcations, which would propagate much faster through the field with respect to the slow front propagation, and push the system through the offset bifurcation (Fig. 6a). To obtain a seizure termination in this way, we need (a) neural dynamics that include several time scales, which are captured by different populations in the Epileptor field model, and (b) that these different time scales are coupled. This coupling across time scales is instantiated in the Epileptor field model by different coupling functions among the populations. This motivates and justifies the coupling function we used: $S(u_1,\theta_{12})$, Eq. 6.

In sum, based on the above arguments, we find two main mechanisms allowing for seizures to terminate synchronously despite their slow spread: (a) the ictal wavefront acts as the main "generator" of ictal activity; when it dies out, the seizure terminates synchronously. However, as we will show in the next section, this scenario cannot account for seizures where the source of SWDs remains stationary as observed in ref. [8]; For this case, a second scenario is required: (b) one of the offset bifurcations propagates much faster than the ictal wavefront, involving the full extent of the recruited neural field. We finally emphasize that our neural field extension of the Epileptor model can account for the cases of not only synchronous but also asynchronous (clustered) seizure termination across different brain regions.

(B) What types of dynamics allow for the two different scenarios of either a moving or a stationary source for ictal spike-and-wave discharges? From a dynamical systems perspective, the number of possible mechanisms is limited. Traveling waves can appear through three distinct mechanisms[23]: (i) A common driver excites different parts of the field with different time delays, eliciting an apparent wave motion. This mechanism is unlikely to be the one responsible for the propagation of ictal SWDs, as it would require a precise organization of time delays between the driver and the different recruited regions of the field; (ii) Propagation of a pulse in an excitable medium or network (Supplementary Fig. 11b). This is the necessary mechanism when considering the scenario of a moving pulse for seizure propagation. Based on this mechanism, as stated in the previous section, the source of the SWDs is then at the location of the slow propagating ictal pulse. Although it can account for the data observed in ref. [12], it cannot account for the cases of stationary source location for SWDs as reported in ref. [8]. As also noted above, there is an exception to this statement if the neural field generating SWDs exhibits hysteresis (Supplementary Fig. 11c). However, we consider this scenario unlikely, because the neural field generating SWDs would always be in a bistable state, even during interictal periods. Consequently, one would be able to generate sustained SWDs at any time with one single stimulation pulse in a region of the brain, which is not commonly observed experimentally. (iii) Coupled-oscillator systems, where phase reorganization can lead to traveling waves. A dynamic phase reorganization could lead to a moving source for the SWDs throughout the seizure (Supplementary Fig. 12c).

These two mechanisms, propagation in excitable media and coupled-oscillator dynamics, can be combined so that the different sites in the field transition into sustained oscillators, i.e., switch from 'resting' (interictal) to ictal oscillatory dynamics. This coupled-oscillator activity occurs in the recruited regions, while the propagation through the excitable media can occur ahead of the ictal wavefront or immediately after the seizure (Fig. 2 and Supplementary Fig. 13). Both mechanisms require a coupling function between the different sites in the field, which justify the introduction of the coupling function $S(u_2,\theta_{22})$, Eq. 10.

**Data availability**. A 100-s sample of the data to reproduce Fig. 1 and the simulation code to reproduce Fig. 2 are available online at the Brown Digital Repository,

https://doi.org/10.7301/Z0PV6HK0. Additional code will be made available upon request by the corresponding author W.T. The full patient datasets cannot be made publicly available as they could compromise the privacy and consent of the participating patients in this study.

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

# ARTICLE

27. Ullah, G., Cressman, J. R. Jr., Barreto, E. & Schiff, S. J. The influence of sodium and potassium dynamics on excitability, seizures, and the stability of persistent states: II. Network and glial dynamics. *J. Comput. Neurosci.* **26**, 171–183 (2009).

28. Ho, E. C. Y. & Truccolo, W. Interaction between synaptic inhibition and glial-potassium dynamics leads to diverse seizure transition modes in biophysical models of human focal seizures. *J. Comput. Neurosci.* **41**, 225–244 (2016).

29. Zhao, M. et al. Preictal and ictal neurovascular and metabolic coupling surrounding a seizure focus. *J. Neurosci.* **31**, 13292–13300 (2011).

30. Wei, Y., Ullah, G. & Schiff, S. J. Unification of neuronal spikes, seizures, and spreading depression. *J. Neurosci.* **34**, 11733–11743 (2014).

31. Suh, M., Ma, H., Zhao, M., Sharif, S. & Schwartz, T. H. Neurovascular coupling and oximetry during epileptic events. *Mol. Neurobiol.* **33**, 181–197 (2006).

32. Wei, Y., Ullah, G., Ingram, J. & Schiff, S. J. Oxygen and seizure dynamics: II. Computational modeling. *J. Neurophysiol.* **112**, 213–223 (2014).

33. Haken, H. *Advanced Synergetics: Instability Hierarchies of Self-Organizing Systems and Devices.* 2nd edn (Springer, Berlin, 1987).

34. Avoli, M. et al. Specific imbalance of excitatory/inhibitory signaling establishes seizure onset pattern in temporal lobe epilepsy. *J. Neurophysiol.* **115**, 3229–3237 (2016).

35. Trevelyan, A. J., Sussillo, D. & Yuste, R. Feedforward inhibition contributes to the control of epileptiform propagation speed. *J. Neurosci.* **27**, 3383–3387 (2007).

36. Jirsa, V. K. & Kelso, J. S. Spatiotemporal pattern formation in neural systems with heterogeneous connection topologies. *Phys. Rev. E* **62**, 8462–8465 (2000).

37. Jirsa, V. K. Neural field dynamics with local and global connectivity and time delay. *Philos. Trans. R. Soc. A: Math., Phys. Eng. Sci.* **367**, 1131–1143 (2009).

38. Saggio, M. L., Spiegler, A., Bernard, C. & Jirsa, V. K. Fast-slow bursters in the unfolding of a high codimension singularity and the ultra-slow transitions of classes. *J. Math. Neurosci.* **7**, 7 (2017).

39. Sanz-Leon, P., Knock, S. A., Spiegler, A. & Jirsa, V. K. Mathematical framework for large-scale brain network modeling in the virtual brain. *NeuroImage* **111**, 385–430 (2015).

40. Spiegler, A., Hansen, E. C. A., Bernard, C., McIntosh, A. R., and Jirsa, V. K. Selective activation of resting-state networks following focal stimulation in a connectome-based network model of the humanBrain. *eNeuro* **3**, e0068–16.2016 (2016).

41. Sanz Leon, P. et al. The Virtual Brain: a simulator of primate brain network dynamics. *Front. Neuroinf.* **7**, 10 (2013).

42. Proix, T., Bartolomei, F., Chauvel, P., Bernard, C. & Jirsa, V. K. Permittivity coupling across brain regions determines seizure recruitment in partial epilepsy. *J. Neurosci.* **34**, 15009–15021 (2014).

43. Durand, D. M., Park, E.-H. & Jensen, A. L. Potassium diffusive coupling in neural networks. *Philos. Trans. R. Soc. B: Biol. Sci.* **365**, 2347–2362 (2010).

44. Amzica, F., Massimini, M. & Manfridi, A. Spatial buffering during slow and paroxysmal sleep oscillations in cortical networks of glial cells in vivo. *J. Neurosci.* **22**, 1042–1053 (2002).

45. Khambhati, A. N., Davis, K. A., Lucas, T. H., Litt, B. & Bassett, D. S. Virtual cortical resection reveals push-pull network control preceding seizure evolution. *Neuron* **91**, 1170–1182 (2016).

46. Jbabdi, S. & Johansen-Berg, H. Tractography: where do we go from here? *Brain Connect.* **1**, 169–183 (2011).

47. Smith, R. E., Tournier, J.-D., Calamante, F. & Connelly, A. The effects of SIFT on the reproducibility and biological accuracy of the structural connectome. *NeuroImage* **104**, 253–265 (2015).

48. Park, Y. S., Hochberg, L. R., Eskandar, E. N., Cash, S. S. & Truccolo, W. Early detection of human focal seizures based on cortical multiunit activity. In *36th Annual International Conference of the IEEE on Engineering in Medicine and Biology Society*, 5796–5799 (2014).

49. Naze, S., Bernard, C. & Jirsa, V. Computational modeling of seizure dynamics using coupled neuronal networks: factors shaping epileptiform activity. *PLOS Comput. Biol.* **11**, e1004209 (2015).

50. Truccolo, W. et al. Single-neuron dynamics in human focal epilepsy. *Nat. Neurosci.* **14**, 635–641 (2011).

51. Uva, L. et al. A novel focal seizure pattern generated in superficial layers of the olfactory cortex. *J. Neurosci.* **37**, 3544–3554 (2017).

52. Jirsa, V., et al. The virtual epileptic patient: individualized whole-brain models of epilepsy spread. *NeuroImage* **145**, 377–388 (2017).

53. Proix, T., et al. How do parcellation size and short-range connectivity affect dynamics in large-scale brain network models? *NeuroImage* **142**, 135–149 (2016).

54. Zalesky, A. et al. Whole-brain anatomical networks: does the choice of nodes matter? *NeuroImage* **50**, 970–983 (2010).

55. El Houssaini, K., Ivanov, A. I., Bernard, C. & Jirsa, V. K. Seizures, refractory status epilepticus, and depolarization block as endogenous brain activities. *Phys. Rev. E* **91**, 010701 (2015).

56. Coombes, S., Lord, G. & Owen, M. Waves and bumps in neuronal networks with axo-dendritic synaptic interactions. *Phys. D: Nonlinear Phenom.* **178**, 219–241 (2003).

57. Izhikevich, E. M. Neural excitability, spiking and bursting. *Int. J. Bifurc. Chaos* **10**, 1171–1266 (2000).

58. Jirsa, V. K., Jantzen, K. J., Fuchs, A. & Kelso, J. S. Spatiotemporal forward solution of the eeg and meg using network modeling. *IEEE Trans. Med. Imaging* **21**, 493–504 (2002).

59. Jirsa, V. K. & Ding, M. Will a large complex system with time delays be stable? *Phys. Rev. Lett.* **93**, 070602 (2004).

60. Feng, J., Jirsa, V. K. & Ding, M. Synchronization in networks with random interactions: theory and applications. *Chaos: Interdisc. J. Nonlinear Sci.* **16**, 015109 (2006).

61. Qubbaj, M. R. & Jirsa, V. K. Neural field dynamics with heterogeneous connection topology. *Phys. Rev. Lett.* **98**, 238102 (2007).

62. Petkoski, S. et al. Heterogeneity of time delays determines synchronization of coupled oscillators. *Phys. Rev. E* **94**, 012209 (2016).

## Acknowledgements

We acknowledge support from the National Institute of Neurological Disorders and Stroke (NINDS), grant R01NS079533 (W.T.); the U.S. Department of Veterans Affairs, Merit Review Award I01RX000668 (W.T.); the Pablo J. Salame '88 Goldman Sachs endowed Assistant Professorship of Computational Neuroscience at Brown University (W.T.). This project received funding from the European Union's Horizon 2020 research and innovation programme under Grant Agreement No. 720270.

## Author contributions

T.P., V.K.J., and W.T. designed research; T.P. performed research; F.B. and M.G. acquired the data; and T.P., V.K.J., F.B., M.G., and W.T. wrote the paper.

## Additional information

**Competing interests:** The authors declare no competing financial interest.

