## [Peer Review File · Nature Communications]

Reviewers' comments:

Reviewer #1 (Remarks to the Author):

There is a marvelous essay in Nature (Nature 427, 297, 2004) by Freeman Dyson describing how Enrico Fermi destroyed years of calculations he had made when despite superb numerical agreement with experimental measurement, the theory was neither based on solid physical underpinnings, nor fundamental mathematical proof (this is the setting of the apocryphal statement from John Von Neumann about 4 and 5 parameters animating an elephant ...).

Reading this present paper drove this reviewer to considerable distress over the seemingly detachment of the neural field extension of the Epileptor neural mass model from the underlying biophysics of the brain elements involved in seizures, and the apparent arbitrariness of the variety of parameter settings. Then I landed disheveled on the shore of the discussion, and it was like listening to insightful masters discuss how epileptic seizures should be approached and understood.

The original Epileptor model in Brain in 2014 was a fascinating paper. It looked at the taxonomy of seizure dynamics, and created a model that could replicate the phenomenological bifurcation types that appeared to occur in experimental data. I felt that there was much value to be gained by organizing a typology of such bifurcations, since they have normal forms, and there was an irreducible underlying set of dynamics that should form a basis for these seizure phenomena.

This present paper extends the neural mass (point) model, to construct spatio-temporal field equations of sufficient complexity to account for features in seizure dynamics which at present we have no proper way to model. There are conflicting papers by excellent experimental groups showing that in human seizures, the dynamics might emanate from stationary or moving fronts, each apparently related to seizure propagation and termination. Clearly seizures can in some patients widely terminate synchronously, and in others terminate with heterogeneous clustering. Often seizures terminate with quasi-periodic spike wave discharges or bursting, and it is these particular types that are studied here.

My principal difficulty with this present work is that I am not convinced that the neural field model represents the underlying physiology with sufficient fidelity to help us uncover some of the missing science we need to gain fundamental understanding of seizure dynamics, nor that the present assumptions represent accurate physiology. I am a bit hampered in this assessment because the modeling as detailed in the methods, and the supplement, are so difficult to follow that I honestly only grasp a fraction of the mathematical embodiment. I will grant the authors that I might be lacking the intellectual capability required, but I think that I represent the intended audience for this present work. I could neither properly understand each component of the formulations, nor could I readily replicate this without being given their present code. Although this could be partially addressed with a substantially expanded and re-written methods and supplement, I feel the need to rate the priority of the present paper low for a broad based general scientific forum, and advocate

that it belongs in a specialty journal for the relevant computational neuroscience or biophysics.

Moderate issues:

Some of the detachment from the neural field model with the experimental data comes from the stereo-EEG (never defined along with other abbreviations and jargon throughout the paper). In such multicontact electrodes orthogonally placed with respect to the normal of the brain, there is no clear relationship between the cortical regions that the field model represents, and the recording sites which cross cortical and white matter boundaries with impunity. One could try to map the cortically transgressing electrodes to an unfolded cortical map, but this was not attempted. At least with traditional subdural electrodes the recordings are all proximal to the crowns of cortical gyri. In this case, it seems much more complex and would have a very variable relationship to the fast and slow field variables.

The relationship between temporal correlations and termination of seizures with spatial coupling via structural and diffusion MRI was quite interesting. The finding that these correlations were consistent with each other seemed a bit obvious – the only surprising thing would have been if they had not been consistent, in which case the findings would have seemed unexplainable. Nevertheless, I very much appreciated the effort here, and this is a nice piece of the paper.

I am not sure that this present model can reconcile what appears to be contradictory experimental findings in refs 6 and 10. I do agree with the authors that the present model could account for such differing findings. But I do not see why the dynamical features of these different recorded human seizures seem qualitatively different to start with, and this has not yet been reconciled in the published literature.

One could claim that my thinking is too '20th Century'. That empirical model fitting to high dimensional data can be considered the 'model', and that the reductionist approach I am advocating with transparent linkages to biophysics is less relevant now. I can see the author's suggestion that the present approach could be used for more patients with heterogeneous seizure patterns, and might lead to improved control strategies. But this would also imply that the present neural field model is irrelevant if one could simply substitute one of comparable dynamical repertoire and the same or increased complexity. However, in a setting where all of the currently available strategies for seizure control could be considered to be outcome failures (vagal nerve, anterior thalamic nucleus, and the responsive neural stimulator), I feel I have license to claim that there has been no substitute for better fundamental understanding and embodiment of that science into models.

Minor issues:

All technical jargon (e.g. Homoclinic bifurcation) and abbreviations (e.g. SEEG, SIFT, ACT, SNIC) need to be defined at first use.

There are many sentences with awkward grammar –
82
112
157
189-190
285
677
705
SI: 45

Does SI: 12 refer to figure S2?

I hope that my comments are helpful and constructive for the authors.

Steven Schiff

Reviewer #2 (Remarks to the Author):

Proix et al. develop an important extension of the Epileptor field model to describe a range of disparate properties seen in human focal epilepsy. The major contribution of the work is that they have put together a unified framework that describes multiple different features of seizure dynamics that occur on different temporal and spatial time scales. While the bulk of the work is simulation based, they do use data from 13 patients to illustrate some of their model descriptions.

Overall, the science and model development are compelling. The work fills a major gap in attempts to provide accurate dynamical system descriptors of epilepsy dynamics.

I have three principle concerns.

1. The authors describe in the Materials and Methods how they extend the Epileptor model to arrive at their current formulation. Perhaps I missed it, but it was not clear to me how the data from the 13 subjects was used to find the model parameters. Could the authors say a little bit more about how the parameters were tuned? Also how robust are the choices of parameters? Are there specific physiological constraints that the authors are imposing to set certain ones of the parameters in specific ranges?

2. The authors show model predictions in Fig. 6 and real data analyses in Fig. 7. There are some qualitative similarities. However, the graphs look appreciable different. Can the authors quantify the degree of agreement between their model predictions and the dynamics seen in actual patient recordings? Quantitative predictions would be preferred in order to eventually make clinical use of the model as the authors suggest in the last paragraph of the Discussion.

3. The findings are quite interesting. However, in many places the writing is awkward and needs editing.

Small issue:

Could the authors define permittivity?

Reviewer #3 (Remarks to the Author):

I enjoyed reading this article, it was fascinating to see the Epileptor model extended to a neural field model with the model predictions confirmed by EEG. This novel work complements the previous extension of the Epileptor model which were coupled via the slow variable. Here the Epileptor fields are coupled via the fast variable, the model shows the link between SWD propagation, connection strength and seizure termination delay. Which are also shown in patient SEEG data. The model is also able to reproduce the stationary or moving source of SWD, that have previously been observed.

The manuscript is valid, well written and well presented. The use of statistics is appropriate. The conclusions are supported by the work. Although the size of the fields used and some of the spatial parameters are only shown graphically, as the simulation code will be available it should be easy to replicate the results. I recommend the manuscript be published.

There are a few minor ambiguities that could easily be corrected;

Fig 1: No y-scale bar

Fig 3: Typo in caption $\gamma_{₁}$

Fig 5B: It is not clear to me what determines the length and location of K maxima (black bars). e.g. $x_{₀} = -1.9$ appears below and to the right of the peak of the K-curve (dashed line) corresponding to $x_{₀} = -1.9$. Could you explain this in the caption or the text.

Line 342: "is in average 2.3 +/- 0.4 longer" No units, are they 2.3 times longer?

Fig 6: How many clusters are there? In B it looks like two clusters, in C and D like 4. Are the green dots in the black box in a different cluster than the green dots in the red box? If so, please use a different color.

Line 390-395: To quantify if the ability of SWDs to propagate ... (see SI Text for details)."
This is not mentioned in the SI text.

Line 483-442: "we processed structural and diffusion MRI data ... (SI Text)." Processing MRI data is not mentioned in the SI Text. Please discuss the difference between the inferred

tractography connection strength and the connection strength in the model, particularly the large difference in scale.

Line 614: "mean age x , range x " replace x .

Eq 3: Should $x_{0,j}$ be x_0 ?

I would also suggest using different names for variables in the equations. For example, x is the distance and x_0 is the excitability parameter, w_1 is the connectivity kernel in (SI Eq 1) but not in (SI Eq 2).

NCOMMS-17-16375: RESPONSE TO REVIEWERS' COMMENTS

Predicting the spatiotemporal diversity of seizure propagation and termination in human focal epilepsy. T Proix, VK Jirsa, F Bartolomei, M Guye, W Truccolo.

REVIEWERS' COMMENTS:**REVIEWER #1:**

1. There is a marvelous essay in Nature (Nature 427, 297, 2004) by Freeman Dyson describing how Enrico Fermi destroyed years of calculations he had made when despite superb numerical agreement with experimental measurement, the theory was neither based on solid physical underpinnings, nor fundamental mathematical proof (this is the setting of the apocryphal statement from John Von Neumann about 4 and 5 parameters animating an elephant ...).

Reading this present paper drove this reviewer to considerable distress over the seemingly detachment of the neural field extension of the Epileptor neural mass model from the underlying biophysics of the brain elements involved in seizures, and the apparent arbitrariness of the variety of parameter settings. Then I landed disheveled on the shore of the discussion, and it was like listening to insightful masters discuss how epileptic seizures should be approached and understood.

The original Epileptor model in Brain in 2014 was a fascinating paper. It looked at the taxonomy of seizure dynamics, and created a model that could replicate the phenomenological bifurcation types that appeared to occur in experimental data. I felt that there was much value to be gained by organizing a typology of such bifurcations, since they have normal forms, and there was an irreducible underlying set of dynamics that should form a basis for these seizure phenomena.

This present paper extends the neural mass (point) model, to construct spatio-temporal field equations of sufficient complexity to account for features in seizure dynamics which at present we have no proper way to model. There are conflicting papers by excellent experimental groups showing that in human seizures, the dynamics might emanate from stationary or moving fronts, each apparently related to seizure propagation and termination. Clearly seizures can in some patients widely terminate synchronously, and in others terminate with heterogeneous clustering. Often seizures terminate with quasi-periodic spike wave discharges or bursting, and it is these particular types that are studied here. My principal difficulty with this present work is that I am not convinced that the neural field model represents the underlying physiology with sufficient fidelity to help us uncover some of the missing science we need to gain fundamental understanding of seizure dynamics, nor that the present assumptions represent accurate physiology.

We thank the reviewer for the important constructive comments and for his appreciation of the earlier work introducing the Epileptor model and a taxonomy of seizure dynamics (Jirsa et al., Brain, 2014) based on bifurcation normal forms for seizure initiation and termination. The reviewer essentially touches upon two important points: 1) how generic is the approach and 2) how physiologically valid is it? We will address both issues separately, but acknowledge that our manuscript positions itself right in between the two extremes.

As expected, the development of an explicit taxonomy based on dynamical system theory for both spatial and temporal aspects of seizure dynamics is a challenging problem. Canonical equations and normal forms exist for ODEs, and normal form theory is well developed using a variety of approaches. For the original Epileptor neural mass model, the authors followed initially the conventions used by Izhikevich (Int J Bifurc Chaos, 2000), and more recently used the more general normal form and unfolding theory (Saggio et al., 2017, J Math Neuroscience). Normal forms are of importance in the qualitative theory of differential equations. Using the theory of singularities of differentiable mappings one can determine which number of terms of the normal form is sufficient to describe the bifurcation of stationary points and periodic solutions up to topological equivalence. Typically, the more generic the analysis is, the fewer the captured data features. This point is evident from Saggio et al. (2017), where the authors used normal form theory to unfold the (Co-dimension 3) singularity of

the Bogdanov-Takens point and identify the topology of qualitative behaviors in its neighborhood, but focused only on the first Epileptor population (seizure onset, offset and slow-variable dynamics),

The extension of canonical approaches to the spatial domain using neural field equations is mathematically significantly less developed. Tools and techniques, well established in the finite dimensional world of dynamical systems (ODEs), do not trivially extend to infinite dimensional evolution equations (PDEs). There are some notable exceptions such as the Birkhoff normal form of Hamiltonian systems, allowing us to view the nonlinear evolution equations in the neighborhood of a stationary solution as small perturbations of infinite dimensional integrable ODEs. Generally, however, when dealing with spatiotemporal evolution equations, these are constrained to providing systematic expansions of the spatial and temporal operators, as well as the polynomial terms. These then lead to basic or fundamental equations, for which however their canonical nature in the sense of normal form theory is not rigorously justified. Examples include fundamental equations that have been investigated for decades including the linear Ginzburg-Landau equations, as well as the famous nonlinear Schroedinger equation, Burgers equation, Korteweg de Vries equations and several others. These equations are fundamental, but they are not canonical in the sense of normal form theory in ODEs.

Having said all this, we think that the two critical aspects for the discussion of our neural field extension of the Epileptor neural mass model are the following: (a) the introduction and treatment of short and long-range spatial connectivity kernels, and (b) the parameter specification for both the coupling strengths for the spatial connectivity and for the nonlinear firing rate function. Specifically:

(a) Regarding the spatial connectivity kernels, there are two fiber systems to be considered: intra- and cortico-cortical fibers (Jirsa, *Phil. Trans. R. Soc. A* 2009 367, 1131-1143). Most of the neural field literature is concerned with intra-cortical connectivity, which is captured by translationally invariant integral kernels (i.e. homogeneous connectivity). To be integrable, the kernels must decay fast enough over space, and are often represented as exponentially decaying functions (Ermentrout and McLeod, 1992, Jirsa & Haken 1996, Jirsa, 2009). These choices are not arbitrary, but informed by statistical analyses of detailed anatomical data from lateral short-range connectivity in the mouse (Braitenberg & Schüz, 1991, Anatomy of the Cortex – Statistics and Geometry) and their extrapolation to the human (Nunez, 1995).

Beyond integrability, approximations based on expansions can further simplify the specification of spatial kernels. Here, we focus on first-order and second-order expansions, which results in exponential spatial kernels (i.e. $\sim \exp(-|x|)$) for intracortical connectivity kernels as a first approximation. The relationship between the expansion terms and the integral kernel for exponential kernels is captured by the equations (2.9) and (2.11) in Jirsa (2009). Solutions of neural field models based on these types of kernels have been shown to typically yield travelling front and wave dynamics (Coombes, 2005). Excitatory on / inhibitory off (e.g. “Mexican hat”) organization that results from higher order expansions of connectivity kernels are thus not included in our Epileptor neural field extension. Nevertheless, inhibitory effects are captured in our model by connectivity already present in the original Epileptor ensemble equations. Our treatment is thus systematic in the sense that there is an approximation of connectivity with truncation criteria and an error term, where the latter converges to zero as the order of the expansion increases. Naturally, there are also choices to be made regarding through which state variables the short-range coupling is to be established. As stated above, these choices can be informed by anatomical/ physiological data (e.g. Braitenberg & Schüz, 1991) and dynamics.

The second fiber system comprises the long-range cortico-cortical fibers and is translationally variant in space (heterogeneous connectivity). This system cannot be approximated locally by definition and needs to be tackled by empirically informed approaches, such as connectomes. To incorporate these long-range cortico-cortical effects in our Epileptor neural field extension, our choice of connectivity was informed by known long-range fiber systems estimated from tractography.

In summary, we treat connectivity and its properties as systematically as possible given the current state of knowledge in pattern-forming systems and brain anatomy/physiology. We concede that the treatment is not as canonical as previous treatments of neural mass models, but we think it represents the best compromise of generic and anatomically/physiologically informed approaches. We now systematically develop this line of

thought in the revised manuscript. Nature Communications allows for a Supplementary Discussion section, which we have now used, together with the revised main text, to convey the approach describe above for the determination of spatial kernels.

(b) Regarding the coupling and nonlinear firing rate parameters: the reviewer's statement "*construct spatio-temporal field equations of sufficient complexity to account for features in seizure dynamics*" also raises the issue of whether our results are more than just fine tuning of a complex model to fit observed data. In particular, reviewer #2 (comment #1) asked about the robustness of the dynamics to different choices of parameters. We performed extensive parameter space explorations and have now reported them in the manuscript (section "Epileptor field model" in the Results and Supplementary Fig. 1; please see also the response to reviewer #2, comment #1). Overall, the analysis reveals that our findings are robust to a large range of parameter variation indicating that these findings are more than simply fine tuning of a complex model.

Finally, we also stress that our neural field extension of the original Epileptor model and the accompanying analyses do indicate a restricted set of possible general cases for the observed spatiotemporal dynamics of seizure propagation and termination. In this sense, our manuscript provides new insights into the spatio-temporal dynamics addressing three main questions: (1) How can seizures often terminate synchronously across large brain regions given that their propagation across these regions is orders of magnitude slower? (2) How can the source of spike-and-wave discharges during seizures move with the slow ictal wavefront in some cases but in others stay stationary at the seizure onset area? and (3) What dynamical mechanisms can contribute to slowing down the ictal wavefront? We have now elaborated in more detail on the contributions related to 3 above points. To keep the response concise, we do not reproduce the corresponding text here. Please refer to the new Supplementary Discussion and the now revised section "Fast oscillations at onset hamper seizure propagation" in the Results section.

In sum, we have revised the manuscript to better convey the above points as well as to clarify the choices made in the neural field extension of the Epileptor model. In addition, we have now included new supplementary figures to add intuition (Supplementary Figs. 8, 9, and 10).

2. I am a bit hampered in this assessment because the modeling as detailed in the methods, and the supplement, are so difficult to follow that I honestly only grasp a fraction of the mathematical embodiment. I will grant the authors that I might be lacking the intellectual capability required, but I think that I represent the intended audience for this present work. I could neither properly understand each component of the formulations, nor could I readily replicate this without being given their present code. Although this could be partially addressed with a substantially expanded and re-written methods and supplement, I feel the need to rate the priority of the present paper low for a broad based general scientific forum, and advocate that it belongs in a specialty journal for the relevant computational neuroscience or biophysics.

We have now revised the manuscript to better describe the modeling work and the corresponding findings, including an expansion of the Methods and supplementary information sections, as well as new schematic figures illustrating the main dynamical system concepts supporting our results for seizure propagation and termination (Supp. Figs. 8-10). Also, we have now made the code and data to reproduce Figs. 1 and 2 available at <https://repository.library.brown.edu/studio/item/bdr:737828/>. This code and sample data will be made publicly available on a digital repository with an assigned DOI. Additional code will be made available upon request to the corresponding author. We hope our responses and revisions have addressed reviewer's concerns.

Moderate issues:

3. Some of the detachment from the neural field model with the experimental data comes from the stereo-EEG (never defined along with other abbreviations and jargon throughout the paper). In such multicontact electrodes orthogonally placed with respect to the normal of the brain, there is no clear relationship between the cortical regions that the field model represents, and the recording sites which cross cortical and white

matter boundaries with impunity. One could try to map the cortically transgressing electrodes to an unfolded cortical map, but this was not attempted. At least with traditional subdural electrodes the recordings are all proximal to the crowns of cortical gyri. In this case, it seems much more complex and would have a very variable relationship to the fast and slow field variables.

We thank the reviewer for raising this issue. We reexamined our data with more stringent criteria to determine which recording sites are included in our analysis. Although the electrodes are indeed orthogonal to the cortical surface, the fact that they are many (between 8 and 13 electrodes, length between 35 and 52 mm) allowed us to span to some extent some of the spatial dimension of the cortex for different structures. In our new analysis, we only took into consideration recording sites that were either in the gray cortical matter or in the hippocampus/amygdala. Additionally, we performed pairwise correlation only between recording contacts belonging to different stereotactic electrodes, therefore avoiding the transverse electrode issue. Overall our results remain qualitatively the same. The alternative of examining seizure termination in recorded ECoG grids is complicated by the fact that typically the grid covers a small surface of adjacent areas where seizures tend to terminate synchronously, thus not providing the important counter-examples for asynchronous (clustered) seizure termination.

4. The relationship between temporal correlations and termination of seizures with spatial coupling via structural and diffusion MRI was quite interesting. The finding that these correlations were consistent with each other seemed a bit obvious – the only surprising thing would have been if they had not been consistent, in which case the findings would have seemed unexplainable. Nevertheless, I very much appreciated the effort here, and this is a nice piece of the paper.

We thank the reviewer for the encouraging comment. We also emphasize that although it might appear obvious that temporal correlations and seizure termination are associated with existing structural connections, the point we are making goes beyond that. We show that seizures tend to end synchronously in regions embedded in functional networks involving correlated activity on fast time scales, i.e. on the time scale of spike-wave discharges. While this finding does not prove causation, it does demonstrate the association of the two aspects (synchronous termination and correlation in fast time scales). We think this is not obvious a priori as activity in this fast time scale could very well be unsynchronized before synchronous seizure termination events. We do find this alternative possibility in the case of ictal activity in two different fields where the seizure might end synchronously, but fast activity is asynchronized. Nevertheless, we do find that long termination delays (clustered/asynchronous seizure termination) are consistently accompanied by low correlation among activity in the fast time scale.

We also note that we have now included a new piece of related analysis as suggested by reviewer 2 (comment #2).

5. I am not sure that this present model can reconcile what appears to be contradictory experimental findings in refs 6 and 10. I do agree with the authors that the present model could account for such differing findings. But I do not see why the dynamical features of these different recorded human seizures seem qualitatively different to start with, and this has not yet been reconciled in the published literature.

As we discuss above in the response to the reviewer's main comment, we think that two different dynamical mechanisms are required for these two types of scenarios. We have now clarified this point in the Discussion and the Supplementary Discussion.

6. One could claim that my thinking is too '20th Century'. That empirical model fitting to high dimensional data can be considered the 'model', and that the reductionist approach I am advocating with transparent linkages to biophysics is less relevant now. I can see the author's suggestion that the present approach could be used for more patients with heterogeneous seizure patterns, and might lead to improved control strategies. But this would also imply that the present neural field model is irrelevant if one could simply substitute one of comparable dynamical repertoire and the same or increased complexity. However, in a setting where all of the currently available strategies for seizure control could be considered to be outcome failures (vagal nerve,

anterior thalamic nucleus, and the responsive neural stimulator), I feel I have license to claim that there has been no substitute for better fundamental understanding and embodiment of that science into models.

Please see our response to comment #1 above.

Minor issues:

7. All technical jargon (e.g. Homoclinic bifurcation) and abbreviations (e.g. SEEG, SIFT, ACT, SNIC) need to be defined at first use.

Done. Thanks.

8. There are many sentences with awkward grammar –

82

112

157

189-190

285

677

705

SI: 45

We thank the reviewer for the careful reading. A copy editor officer at Brown has now seen the revised manuscript and has helped to improve the writing. We hope this issue has been satisfactorily addressed.

9. Does SI: 12 refer to figure S2?

We could not find “SI: 12” in the original main or supplementary text. Perhaps we misunderstood the reviewer’s question?

10. I hope that my comments are helpful and constructive for the authors.

Steven Schiff

We thank the reviewer for his constructive comments. We hope our revisions have allayed the reviewer’s concerns.

REVIEWER #2:

Proix et al. develop an important extension of the Epileptor field model to describe a range of disparate properties seen in human focal epilepsy. The major contribution of the work is that they have put together a unified framework that describes multiple different features of seizure dynamics that occur on different temporal and spatial time scales. While the bulk of the work is simulation based, they do use data from 13 patients to illustrate some of their model descriptions.

Overall, the science and model development are compelling. The work fills a major gap in attempts to provide accurate dynamical system descriptors of epilepsy dynamics.

We thank the reviewer for the encouraging comments and appreciation of our manuscript.

I have three principle concerns.

1. The authors describe in the Materials and Methods how they extend the Epileptor model to arrive at their current formulation. Perhaps I missed it, but it was not clear to me how the data from the 13 subjects was used to find the model parameters. Could the authors say a little bit more about how the parameters were tuned?

Also how robust are the choices of parameters? Are there specific physiological constraints that the authors are imposing to set certain ones of the parameters in specific ranges?

We thank the reviewer for the valuable suggestions which have led to additional analysis involving extensive parameter space exploration (Supplementary Fig. 1) in the revised manuscript.

We emphasize that the results presented in Figures 6 explain and predict the association between the level of correlation in the fast activity across different sites and whether their ictal activity terminates synchronously or asynchronously. This prediction is then tested in the 13-patient dataset. We clarify that the SEEG data were not used to fit Epileptor field model parameters in order to obtain this prediction. The parameters of the Epileptor neural field were the same as in the original Epileptor neural mass model, except for a multiplicative scalar a_{12} of the coupling of the action of the fast variable u_1 on q_1 (Eq. 6), as stated in the Methods. This parameter change shifts the propagation of spike-wave discharges toward the beginning of the seizure. In addition, in the neural field extension we added three local coupling functions and a heterogeneous coupling function.

Please see our response to Reviewer 1 for the details on the choice of the coupling function. Regarding the coupling parameters, each of the local coupling functions introduces two parameters: θ_{ij} (activity threshold) and γ_{ij} (coupling strength). The activity thresholds were specific to each population, and the coupling strength parameters were set to $\gamma_{11} = \gamma_{22} = \gamma_{12}/10$ without a priori tuning to the data. $\gamma_{11} = \gamma_{22} = \gamma_{12}/10$

To assess if the Epileptor neural field dynamics are robust to variations in these six parameters, we have now added a new piece of analysis based on the systematic exploration of the corresponding parameter space. We also note that variation in the parameters γ_{11} and θ_{11} was already explored in Fig. 3C-D, showing that the system exhibits a propagating front over a large range of value of these two parameters. We added a new figure to explore the effect of the four remaining parameters (new Supplementary Fig. 1). We have now also justified in more detail the choice of these parameters in the Material and Methods.

For the heterogeneous coupling function, we likewise have two parameters, i.e. the coupling strength $\gamma_{het,j}$, and the activity threshold θ_{het} . Variations in the coupling strength parameter $\gamma_{het,j}$ were already systematically examined in Fig. 6D. The main effect is the change for the second Epileptor field model from a regime where the seizure does not recruit the second field to a regime where it does. When the second Epileptor field model is recruited, the strength of the heterogeneous coupling affects the seizure termination delays and the correlation, as shown in Fig 6C-D. The activity threshold parameter θ_{het} has similar effects. We have now revised the text to make this result clearer.

In sum, the qualitative dynamics of the Epileptor field model are robust to these parameter variations. Please see also our answer to reviewer 1's main comment.

2. The authors show model predictions in Fig. 6 and real data analyses in Fig. 7. There are some qualitative similarities. However, the graphs look appreciable different. Can the authors quantify the degree of agreement between their model predictions and the dynamics seen in actual patient recordings? Quantitative predictions would be preferred in order to eventually make clinical use of the model as the authors suggest in the last paragraph of the Discussion.

As stated above, the analysis and simulations of the Epileptor field model predict and explain the association between the level of correlation in the fast activity across different sites and whether their ictal activity terminates synchronously or asynchronously as examined in Figure 6. This predicted association was then confirmed in the 13-patient dataset as shown in Figure 7. Clustering and statistical analysis quantified this association in each case (simulated model and patient data). In this way, our model contributes towards a more realistic modeling of spatio-temporal dynamics of seizure onset, propagation and termination, and is part of a larger effort towards building patient-specific models to assist clinicians' diagnosis and intervention as outlined in Jirsa et al. (NeuroImage, 2016). Also, based on the current work, we hope to develop an approach for predicting whether a seizure will terminate synchronously or in a clustered fashion based on the patient

specific connectivity and the observed neural state at the beginning of the seizure. In order to achieve this, one needs first to predict, using the above mentioned information and model simulation, how the spatial correlation between spike-wave discharges across many different brain sites will evolve as the seizure progresses. (This prediction will vary across different seizures in the same patients, since different seizure may terminate synchronously or asynchronously.) This remains a challenging problem which we hope to address in the future. We have now also added a statement in the Discussion to address this point.

Regarding the differences between the model predictions (Fig. 6) and the patient data (Fig. 7) referred by the reviewer: we note that there are two main differences. First, the distribution of termination delays can be more continuous for the simulated Epileptor fields than in the actual data. By systematically changing the heterogeneous connection strength in the model, it is possible to obtain very small termination delays between two different fields connected by long-range connections. We think this difference between simulation and data is not critical, since it seems likely that the heterogeneous connection strengths in our patient datasets have a much more limited range and a sparse distribution. Perhaps, with larger datasets, a more continuous variation might be observed. We also note the possibility that in some cases, when a seizure terminates in two distinct fields synchronously, one might also find both small pairwise termination delays and low pairwise correlation values between sites located in the two different fields. This is likely the case in the actual data, where potentially there might be multiple interacting fields, providing a possible explanation for the observed case of low correlation values for small termination delays (Fig. 7A).

Second, the distribution of correlation values in Fig. 7A is more continuous than in Fig 6B for low termination delays. This can be explained by both the presence of noise in the actual data, weakening correlation values compared to deterministic simulations, and by the presence of more than one field, as explained above. We have now repeated the same simulations with added noise, which resulted in a more continuous range of correlation values. The revised Fig. 6B shows a better agreement with Figure 7A in this regard.

We have also added a new Fig. S6 to show other examples of simulated and actual seizures and to illustrate the two source of differences as discussed above.

Furthermore, we have now reexamined the patient data to perform the analysis only on those electrodes that were located in gray matter (please see our response to reviewer 1's comment #3). New analyses are also now included to better check our model predictions. Specifically, we have now checked if contact pairs with long termination delays are found in different fields, as predicted by our model. To do so, for each set of contact pairs, we systematically identified the two corresponding brain regions, and assessed if these two regions were contiguous in the brain atlas with a binary score (one if contiguous, zero otherwise). The mean region/contact proximity was then computed for each cluster of pairwise delays (new Fig. 7D). We found a significant difference of contact proximity between cluster one (small termination delays) versus the other clusters (long termination delays; Fig 7E). This confirms that long termination delays tend to occur when contact pairs are in different fields. In the revised figures, the same logarithmic scale is now used for termination delays both in the model and in the data to allow for easier comparison between the two.

3. *The findings are quite interesting. However, in many places the writing is awkward and needs editing.*

As stated above in the response to reviewer 1, a copy editor officer at Brown has now seen the revised manuscript and has helped to improve the writing. We hope this issue has been satisfactorily addressed.

Small issue:

4. *Could the authors define permittivity?*

As in the original Epileptor model, the slow variable in the Epileptor field model is termed "permittivity." The slow variable represents the "distance" of the Epileptor state to the seizure threshold, i.e. permittivity to seizure states or the ability of the model to resist to seizure triggering events (e.g. inputs from other brain regions, stimulation, noise, etc.) The term permittivity was adopted by analogy to "permittivity" in electromagnetism, where it describes the ability of a material to resist to an electric field. We have now revised the text to define

the term permittivity on its first occurrence.

REVIEWER #3:

I enjoyed reading this article, it was fascinating to see the Epileptor model extended to a neural field model with the model predictions confirmed by EEG. This novel work complements the previous extension of the Epileptor model which were coupled via the slow variable. Here the Epileptor fields are coupled via the fast variable, the model shows the link between SWD propagation, connection strength and seizure termination delay. Which are also shown in patient SEEG data. The model is also able to reproduce the stationary or moving source of SWD, that have previously been observed.

The manuscript is valid, well written and well presented. The use of statistics is appropriate. The conclusions are supported by the work. Although the size of the fields used and some of the spatial parameters are only shown graphically, as the simulation code will be available it should be easy to replicate the results. I recommend the manuscript be published.

We thank the reviewer for the encouraging comments.

There are a few minor ambiguities that could easily be corrected;

1. Fig 1: No y-scale bar

We have now added a y-scale bar in Fig. 1.

2. Fig 3: Typo in caption γ_1

Thanks for noticing it. We have now corrected the typo.

3. Fig 5B: It is not clear to me what determines the length and location of K maxima (black bars). e.g. $x_0 = -1.9$ appears below and to the right of the peak of the K-curve (dashed line) corresponding to $x_0 = -1.9$. Could you explain this in the caption or the text.

We apologize for the imprecise phrasing in the caption of Fig. 5. In Fig. 5B, the maxima of K refer to maxima over space and not over time. Therefore, the length and location of K maxima in this plot is determined by choosing at each time point which of the four curves has the highest K value.

4. Line 342: "is in average 2.3 +/- 0.4 longer" No units, are they 2.3 times longer?

The reviewer is correct. The sentence should have read "is in average 2.3 +/- 0.4 times longer". We have now corrected it.

5. Fig 6: How many clusters are there? In B it looks like two clusters, in C and D like 4. Are the green dots in the black box in a different cluster than the green dots in the red box? If so, please use a different color.

We apologize for the lack of clarity in the figure and caption. In Fig. 6B, the clusters are identified by the color of the dots (green and purple dots). While in the example of Fig.6B there are only two clusters, in other seizures there were up to four clusters. The red and black box indicate if the two sites used for computing the pairwise correlation are in the same or a different field. We have now revised Fig. 6B to facilitate the interpretation, and show other examples of clustering in the new Supplementary Fig. 7.

The clusters in Fig 6B are determined based on the termination delay only, as shown by the green dots in Fig. 6B. In the actual data, a large extent of correlation values for short termination delays is found, which indicates the possibility of a seizure terminating synchronously in different fields, thus resulting in low termination delays and low correlation values. This possibility is confirmed by model simulations, where the clustering can give an

incorrect estimation of cluster membership in the model for two sites in the same or a different field (as indicated by the presence of green dots in the red box in the new Supplementary Fig. 7B). Because the exact extension of the fields is unknown, we used termination delays instead. Termination delays can be easily clustered in the SEEG data. Our model therefore accounts for the case of misclassification that could arise from a seizure terminating synchronously in different fields. Based on this clustering, we have now confirmed in more detail that long termination delays between contact pairs systematically imply a low correlation between spike-wave discharges in the corresponding pairs. Please see also the answer to reviewer 2 (comment #2).

6. Line 390-395: *To quantify if the ability of SWDs to propagate ... (see SI Text for details). This is not mentioned in the SI text.*

Indeed, the details are available in Materials and Methods, no in the SI Text. We have now corrected this typo.

7. Line 483-442: *"we processed structural and diffusion MRI data ... (SI Text)." Processing MRI data is not mentioned in the SI Text. Please discuss the difference between the inferred tractography connection strength and the connection strength in the model, particularly the large difference in scale.*

As in the above, the details are available in Materials and Methods, not in SI Text. We have now corrected the typo. The connection strength is here quantified by the number of fiber tracks connecting two regions as inferred from tractography. This measure is only a proxy for the connection strength, as tractography cannot quantify the effective effect of the connection on the local dynamics (Jbabdi et al, 2011). The actual relationship, while likely monotonic, is possibly nonlinear. A suggestion here is that such effect could be logarithmic. However, as our observation is based on the existence of separate clusters and not on the fit of the connection strength parameters in the model to the data, we think this is not an issue. We have also changed the term from "connection strength" to "number of tracks" (revised Fig. 7 and the corresponding caption) to avoid ambiguity, and we now comment on this point in the Discussion.

8. Line 614: *"mean age x, range x" replace x.*

We apologize for this omission. We have now replaced it with the correct values.

9. Eq 3: *Should $x_{0,j}$ be x_0 ?*

Indeed, that was a typo. $x_{\{0, j\}}$ should be $x_0(x)$. We have now corrected it.

10. *I would also suggest using different names for variables in the equations. For example, x is the distance and x_0 is the excitability parameter, w_1 is the connectivity kernel in (SI Eq 1) but not in (SI Eq 2).*

We have now checked and modified the variable names to avoid the use of the same or similar symbol in different contexts.

REVIEWERS' COMMENTS:

Reviewer #1 (Remarks to the Author):

Review revised NCOMMS-17-16375A

This is an extensively revised manuscript. In the original version, I had 3 main concerns: 1) that the model was too complex for the underlying physiology, 2) that the exposition was unclear, and 3) that the SEEG contained uninterpretable measurements that were not cortical for this cortical model.

This is a vastly improved manuscript. The authors have done a good job explaining #1, the manuscript is much clearer now, and now only reasonable SEEG electrode locations were employed.

The results are important and instructive. Given that the literature is quite confusing in its attempt to explain contradictory findings of seizure termination with spike wave discharges (SWD), which seem to at times emanate from a stationary focus, and at times propagate from a shifting focus, we are still left with the universal mystery of the frequent synchronous termination in SWD of many but not all seizures. This manuscript proposes that with a reasonably elemental model, that the bifurcations can travel faster than the ictal wavefront. There is a fundamental interplay between temporal and spatial scales that accounts for this, along with the strength of short and longer range coupling. The model predictions are consistent with findings in SEEG seizures, and in connectome measurements from patients. The insight that the region where stimulation would be effective might be much larger than the apparent focus is both profound and consistent with the growing body of responsive stimulation data. The importance of early response to seizure onset is an important hypothesis to test.

It is my opinion that this work now makes important statements about seizure dynamics, and that my major objections have been well handled. That said, the story is not over. My initial concerns that simulation of phenomenology using a phenomenological model is still removed from the level of fundamental understanding that I hope will eventually emerge in our understanding of these dynamics. The authors understand this, stating towards the end of the discussion: "As a phenomenological model, the Epileptor field model identifies the invariant features that constrain the observed dynamics and may inform the development of detailed biophysical models." I find the lumping of slow permittivity parameters attractive – it certainly does work in this model. But the lack of explicit inhibitory elements is something that seems uncomfortable at best, albeit their nice discussion of this decision. I do appreciate their statement in the supplementary discussion that their present modeling "represents the best compromise of generic and anatomically/physiologically informed approaches." I hope that the future results will find a more fundamental set of models that are validated at the level of the underlying biophysics.

Moderate Recommendation

I would like to recommend that Supplementary Material figure 6 be added to the Main Figure 2 to contrast this important case of non-synchronous termination. I also find that the plot of v is helpful in this supplementary plot.

Minor Issues

[n.b.: all line numbers I refer to here are from the version with Tracking Changes submitted as a 'related manuscript'.]

Lines 31-34: Should not use text and date citations in a manuscript that uses numerically indexed citations. At least give the numerical citations as well so that they can be identified.

Lines 22-44: SWD just sort of appears, and the lines never really make the point as to why. A conversation in progress that a general reader who is not an epilepsy expert will likely be confused about. Please take a few sentences and explain the SWD phenomenon in more detail and why it is important.

Figure 2 legend – insert period after the title.

Figure 3 and 6c,d need color maps with calibration

Line 407 'an' should be 'a'

Figure 7 would be clearer if there were a map of patients #1, 2, ... , n (or use the patient initials as in the Supplementary table) and the color and symbols used. This is more a matter of taste (mine) than the need for calibrated maps as in Figures 3 and 6

Supplementary Figure 1 needs a colorbar with calibration, and remove in the legend the designation of 'low' and 'high' colors.

Steven Schiff

Reviewer #2 (Remarks to the Author):

I have no further comments.

Reviewer #3 (Remarks to the Author):

Thank you for correcting or clarifying the few points I made, they have all been satisfactorily addressed. The additional supplementary discussion and figures provide a beneficial insight into your modeling approach. As before, I recommend the manuscript is published.

NCOMMS-17-16375A: RESPONSE TO REVIEWERS' COMMENTS**Predicting the spatiotemporal diversity of seizure propagation and termination in human focal epilepsy. T Proix, VK Jirsa, F Bartolomei, M Guye, W Truccolo.****REVIEWERS' COMMENTS:****REVIEWER #1:**

This is an extensively revised manuscript. In the original version, I had 3 main concerns: 1) that the model was too complex for the underlying physiology, 2) that the exposition was unclear, and 3) that the SEEG contained uninterpretable measurements that were not cortical for this cortical model.

This is a vastly improved manuscript. The authors have done a good job explaining #1, the manuscript is much clearer now, and now only reasonable SEEG electrode locations were employed.

The results are important and instructive. Given that the literature is quite confusing in its attempt to explain contradictory findings of seizure termination with spike wave discharges (SWD), which seem to at times emanate from a stationary focus, and at times propagate from a shifting focus, we are still left with the universal mystery of the frequent synchronous termination in SWD of many but not all seizures. This manuscript proposes that with a reasonably elemental model, that the bifurcations can travel faster than the ictal wavefront. There is a fundamental interplay between temporal and spatial scales that accounts for this, along with the strength of short and longer range coupling. The model predictions are consistent with findings in SEEG seizures, and in connectome measurements from patients. The insight that the region where stimulation would be effective might be much larger than the apparent focus is both profound and consistent with the growing body of responsive stimulation data. The importance of early response to seizure onset is an important hypothesis to test.

It is my opinion that this work now makes important statements about seizure dynamics, and that my major objections have been well handled. That said, the story is not over. My initial concerns that simulation of phenomenology using a phenomenological model is still removed from the level of fundamental understanding that I hope will eventually emerge in our understanding of these dynamics. The authors understand this, stating towards the end of the discussion: "As a phenomenological model, the Epileptor field model identifies the invariant features that constrain the observed dynamics and may inform the development of detailed biophysical models." I find the lumping of slow permittivity parameters attractive – it certainly does work in this model. But the lack of explicit inhibitory elements is something that seems uncomfortable at best, albeit their nice discussion of this decision. I do appreciate their statement in the supplementary discussion that their present modeling "represents the best compromise of generic and anatomically/physiologically informed approaches." I hope that the future results will find a more fundamental set of models that are validated at the level of the underlying biophysics.

We thank the reviewer for the encouraging and important comments.

Moderate Recommendation:

I would like to recommend that Supplementary Material figure 6 be added to the Main Figure 2 to contrast this important case of non-synchronous termination. I also find that the plot of v is helpful in this supplementary plot.

Done. Thanks for the suggestion.

Minor Issues:

[n.b.: all line numbers I refer to here are from the version with Tracking Changes submitted as a 'related manuscript'.]

Lines 31-34: Should not use text and date citations in a manuscript that uses numerically indexed citations. At least give the numerical citations as well so that they can be identified.

Done.

Lines 22-44: SWD just sort of appears, and the lines never really make the point as to why. A conversation in progress that a general reader who is not an epilepsy expert will likely be confused about. Please take a few sentences and explain the SWD phenomenon in more detail and why it is important.

We have now added the following statements in the Introduction (lines 24 – 33):

Here, we consider focal seizures that start with or evolve into SWDs (15). SWDs during focal seizures are characterized by a large amplitude spike in the field potential followed by a slower wave with the opposite polarity. More complex morphology, e.g. poly-spikes, is also common. SWD events tend to recur 2-3 times per second during a seizure. Typically, neuronal action-potential activity increases during the spike phase and is largely suppressed during the wave phase (16, 17). We focus on two main related aspects of these SWD seizures.

Figure 2 legend – insert period after the title.

Done.

Figure 3 and 6c,d need color maps with calibration

Done. Thanks.

Line 407 'an' should be 'a'

Corrected.

Figure 7 would be clearer if there were a map of patients #1, 2, ... , n (or use the patient initials as in the Supplementary table) and the color and symbols used. This is more a matter of taste (mine) than the need for calibrated maps as in Figures 3 and 6

We attempted to add the corresponding labels for each patient in Figure 7, but we thought the addition added too much clutter to the figure. Since the reviewer stated this suggestion was optional, we decided not to implement it. Nevertheless, we have added calibration maps in the other figures as requested.

Supplementary Figure 1 needs a colorbar with calibration, and remove in the legend the designation of 'low' and 'high' colors.

We have now added colorbars with calibration in Supp. Figs. 1, 4 and 5.

In addition, we also note that, as requested by the editor, all of the supplementary methods and methodological considerations have been moved to the Methods section in the main text. The Supplementary Information document now contains only the Supplementary Figures.

REVIEWER #2:

I have no further comments.

We thank the reviewer for the important previous comments and suggestions.

REVIEWER #3:

Thank you for correcting or clarifying the few points I made, they have all been satisfactorily addressed. The additional supplementary discussion and figures provide a beneficial insight into your modeling approach. As before, I recommend the manuscript is published.

We thank the reviewer for the important previous comments and suggestions.